



# On the Processes Determining the Slope of Cloud-Water Adjustments in Non-Precipitating Stratocumulus

 Fabian Hoffmann[1],  Yao-Sheng Chen[2,3], and  Graham Feingold[3]

[1]Ludwig-Maximilans-Universität München, Meteorologisches Institut, Munich, Germany
[2]Cooperative Institute for Research in Environmental Sciences, University of Colorado Boulder, Boulder, Colorado, USA
[3]Chemical Sciences Laboratory, NOAA, Boulder, Colorado, USA

**Correspondence:** Fabian Hoffmann (fa.hoffmann@lmu.de)

**Abstract.** Cloud-water adjustments are a part of aerosol-cloud interactions, affecting the ability of clouds to reflect shortwave radiation by processes altering the vertically integrated cloud water content $L$ in response to changes in the droplet concentration $N$. In this study, we utilize a simple entrainment parameterization for mixed-layer models to determine entrainment-mediated cloud-water adjustments in non-precipitating stratocumulus. At lower $N$, $L$ decreases due to an increase in entrainment in response to an increase in $N$ suppressing the stabilizing effect of evaporating precipitation (virga) on boundary layer dynamics. At higher $N$, the cessation of cloud-droplet sedimentation sustains more liquid water at the cloud top, and hence stronger preconditioning of free-tropospheric air, which increases entrainment with $N$. Overall, cloud-water adjustments are found to weaken distinctly from $\mathrm{d}\ln(L)/\mathrm{d}\ln(N) = -0.48$ at $N = 100\,\mathrm{cm}^{-3}$ to $-0.03$ at $N = 1000\,\mathrm{cm}^{-3}$, indicating that a single value to describe cloud-water adjustments in non-precipitating clouds is insufficient. Based on these results, we speculate that cloud-water adjustments at lower $N$ are associated with slow changes in boundary layer dynamics, while a faster response is associated with the preconditioning of free-tropospheric air at higher $N$.

## 1 Introduction

By determining the concentration of cloud droplets $N$, aerosol has a substantial effect on the optical properties of clouds and their role in Earth's radiation budget (e.g., Boucher et al., 2013; Forster et al., 2021). While it is well known that an increase in $N$ results in stronger reflection of incident solar shortwave radiation by increasing the number of scatterers (Twomey, 1974, 1977), further changes in the cloud micro- and macrostructure are less well understood. Especially concurrent changes in the vertically integrated cloud water content, the so-called liquid water path $L$, are important to consider, as they strengthen or weaken any impact of the aforementioned *Twomey effect* (e.g., Stevens and Feingold, 2009). These aerosol-mediated changes in $L$ are referred to as cloud-water adjustments and they are often quantified as

$$m \equiv \frac{\mathrm{d}\ln(L)}{\mathrm{d}\ln(N)}. \tag{1}$$

$m$ tends to be positive when an increase in $N$ suppresses precipitation and increases $L$ due to smaller cloud droplets (Albrecht, 1989). In the absence of precipitation, further increases in $N$ are found to increase the mixing of clouds with their surroundings (entrainment), leading to a decrease in $L$ (Wang et al., 2003; Ackerman et al., 2004; Bretherton et al., 2007). Thus, $m$ tends to





be positive for precipitating clouds and negative for non-precipitating clouds (e.g., Gryspeerdt et al., 2019). While the $m$ at low
and high $N$ are likely related (Hoffmann et al., 2024b), the value of $m$ in non-precipitating clouds is not understood, with most
observational estimates ranging from $-0.4$ to $-0.2$ (e.g., Christensen and Stephens, 2011; Gryspeerdt et al., 2019; Possner
et al., 2020), with a suggested lower limit of $-0.64$ (Glassmeier et al., 2021). Moreover, a single $m$ to describe cloud-water
adjustments in non-precipitating stratocumulus might not be sufficient as $m$ seems to be a function of $N$ (e.g., Lu and Seinfeld,
2005; Chen et al., 2011).

This variability of $m$ in non-precipitating clouds is likely associated with a complex network of interactions and dependencies that comprise entrainment (Mellado, 2017; Igel, 2024), making it hard to obtain direct process understanding from three-dimensional modeling, such as large-eddy simulation. Thus, to understand $m$ for non-precipitating stratocumulus better, we
will base our work on a simple, zero-dimensional mixed-layer model (Lilly, 1968; Schubert et al., 1979; Bretherton and Wyant,
1997; Stevens, 2002). We will focus on the representation of the entrainment rate in such models and how it depends on $L$
and $N$ (Nicholls and Turton, 1986; Turton and Nicholls, 1987; Bretherton et al., 2007). Despite this fundamentally simpler
approach, our mixed-layer model results agree qualitatively with large-eddy simulations published in a companion paper (Chen
et al., 2024a).

The study is structured as follows. In Section 2, we will lay out the mathematical framework of the applied model, covering
the fundamentals of the analyzed mixed-layer model entrainment parameterization and the approach to extract $m$ from that
parameterization. Results are presented in Section 3, and conclusions are drawn in Section 4.

## 2 Mathematical Framework

### 2.1 Mixed-Layer Models

For mixed-layer models (e.g., Schubert et al., 1979), it is a convenient choice to describe the boundary layer predicting the
moist static energy

$$s = c_\mathrm{p}T + gh + L_\mathrm{v}q_\mathrm{v}, \tag{2}$$

the total water mixing ratio

$$q_\mathrm{t} = q_\mathrm{v} + q_\mathrm{l}, \tag{3}$$

and the boundary layer depth

$$h_\mathrm{t}. \tag{4}$$

Here, $c_\mathrm{p}$ is the specific heat capacity of air at constant pressure, $T$ absolute temperature, $g$ acceleration by gravity, $h$ height
above surface, $L_\mathrm{v}$ enthalpy of evaporation, $q_\mathrm{v}$ water vapor mixing ratio, and $q_\mathrm{l}$ liquid water mixing ratio.

Changes in $s$ and $q_\mathrm{t}$ are determined by their respective fluxes from the surface, $\overline{w's'}|_0$ and $\overline{w'q_\mathrm{t}'}|_0$, and the free troposphere,
$w_\mathrm{e}\Delta s$ and $w_\mathrm{e}\Delta q_\mathrm{t}$, where $w_\mathrm{e}$ is the entrainment rate and $\Delta s$ and $\Delta q_\mathrm{t}$ are the respective changes (jumps) of $s$ and $q_\mathrm{t}$ from the





mixed layer to the free troposphere. Further, $s$ and $q_t$ are affected by processes taking place inside the mixed layer, such as emission and absorption of radiation affecting $s$ and precipitation reducing $q_t$. $h_t$ is determined by the interplay of $w_e$ and subsidence $w_s$. In this study, however, we will solely focus on how $w_e$ depends on $L$ and $N$, which will give us an estimate of how entrainment affects cloud-water adjustments.

## 2.2 Entrainment

The foundation of many $w_e$ parameterizations is the scaling

$$\frac{w_e}{w_*} \sim \mathrm{Ri}^{-1} \tag{5}$$

by Turner (1973). This relationship states that $w_e$ is proportional to a characteristic velocity of the flow, e.g., the convective velocity $w_*$, with the constant of proportionality being determined by the inverse Richardson number

$$\mathrm{Ri}^{-1} = \frac{w_*^2}{h_t \Delta b}. \tag{6}$$

This scaling reflects that the dynamics of the boundary layer, represented by $w_*$, have to be sufficiently strong to overcome the stably stratified boundary layer top, represented by the buoyancy jump between the boundary layer and free troposphere $\Delta b$, to mix free-tropospheric air into the boundary layer. Since Deardorff (1976) showed that

$$w_*^3 = \frac{2}{1 - k_*} \int_0^{h_t} \overline{w'b'}(h)\, \mathrm{d}h = \frac{2}{1 - k_*} h_t \langle B \rangle, \tag{7}$$

where $k_* = 0.2$ is a factor relating $w_*^3$ to the buoyancy flux $\overline{w'b'}$, it is possible to relate

$$w_e = A \frac{\langle B \rangle}{\Delta b}, \tag{8}$$

introducing the vertically averaged buoyancy flux $\langle B \rangle$ from (7) and the entrainment efficiency $A$. In the following, we will address how $\Delta b$, $\langle B \rangle$, and $A$ are specified. Figure 1 shows all these contributions and the resulting $w_e$ in an $L$-$N$ phase space, with more detailed explanations following in Section 3.

### 2.2.1 The Buoyancy Jump $\Delta b$

Following Schubert et al. (1979), we define the virtual static energy as

$$s_v = s - L_v q_v + c_p T_0 (\eta q_v - q_l) = s - (1 - \eta \epsilon) L_v q_t + [1 - (1 + \eta) \epsilon] L_v q_l, \tag{9}$$

with the definitions $\epsilon = (c_p T_0)/L_v$ and $\eta = R_v/R_d - 1$, with $R_v$ and $R_d$ the specific gas constants of water vapor and dry air, respectively, and $T_0$ a reference temperature. Thus, the buoyancy of a parcel of air is

$$b = \frac{g}{T_0 c_p} s_v, \tag{10}$$



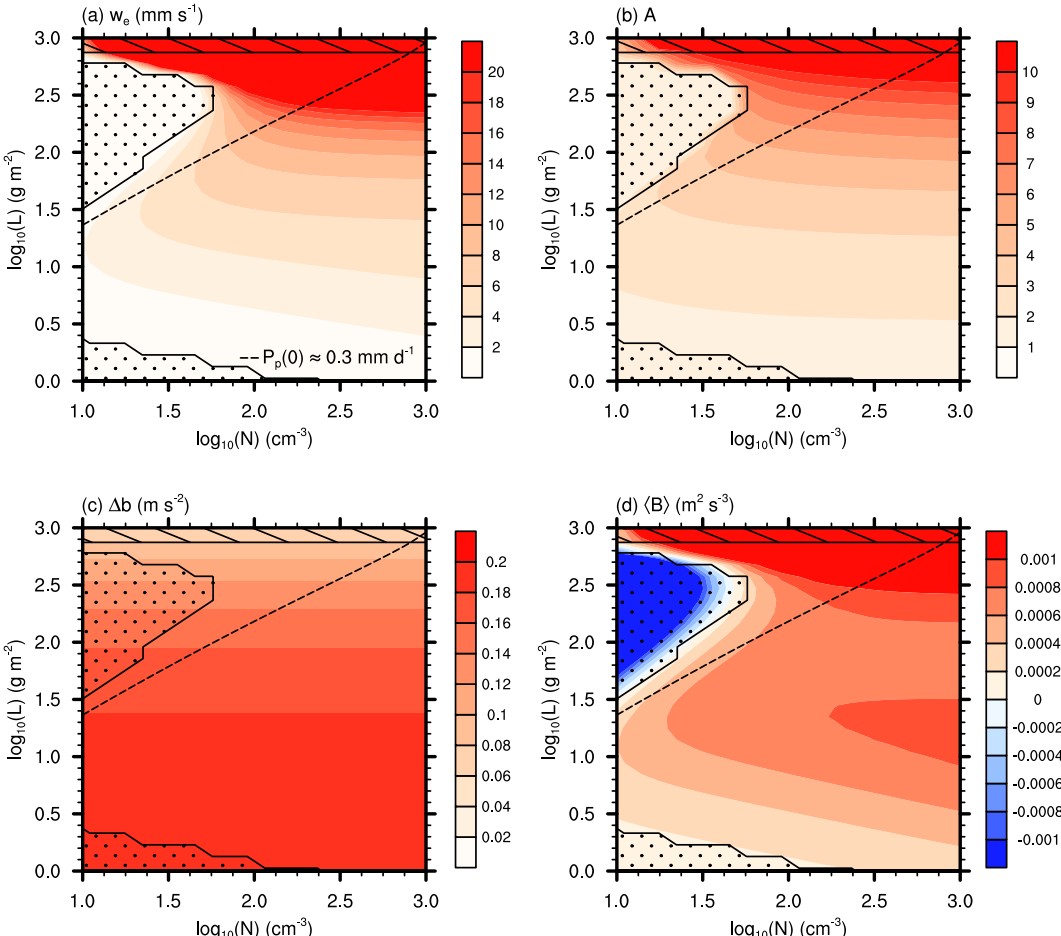

**Figure 1.** A representation of (a) the entrainment velocity $w_e$, (b) entrainment efficiency $A$, (c) buoyancy jump $\Delta b$, and (d) vertically averaged buoyancy flux $\langle B \rangle$ as a function of the liquid water path $L$ and droplet number concentration $N$. Stippled regions denote decoupled boundary layers, while striped regions indicate that the cloud base is at or below the surface. These regions should not be analyzed. The dashed line indicates a surface precipitation rate of $P_p(0) = 0.1 \overline{w'q'_t}|_0 \rho_0 \approx 0.3 \, \mathrm{mm \, d^{-1}}$, separating precipitating (to the left) from non-precipitating (to the right) parts of the phase space.

and the buoyancy jump between the top of the boundary layer to the free troposphere

$$\Delta b = b(h_t + \delta h) - b(h_t) = \frac{g}{T_0 c_p} \left\{ \Delta s - (1 - \eta \epsilon) L_v \Delta q_t - [1 - (1 + \eta) \epsilon] L_v q_l(h_t) \right\}, \tag{11}$$

where $\delta h$ is an (assumed small) height difference from the boundary layer top to the free troposphere, defining the aforementioned jumps $\Delta s = s(h_t + \delta h) - s(h_t)$ and $\Delta q_t = q_t(h_t + \delta h) - q_t(h_t)$. Note that the free-tropospheric $q_l(h_t + \delta h) = 0$ per definition.





### 2.2.2 The Average Buoyancy Flux $\langle B \rangle$

The dynamics of the stratocumulus-topped boundary layer are primarily driven by buoyancy, and the corresponding buoyancy flux can be expressed as

$$\overline{w'b'}(h) = \frac{g}{T_0 c_{\mathrm{p}}} \overline{w's_{\mathrm{v}}'}(h) = \frac{g}{T_0 c_{\mathrm{p}}} \left[ \alpha_{\mathrm{s}}(h) \overline{w's'}(h) - \alpha_{\mathrm{q_t}}(h) L_{\mathrm{v}} \overline{w'q_{\mathrm{t}}'}(h) \right], \tag{12}$$

where the coefficients

$$\alpha_{\mathrm{s}}(h) = \begin{cases} 1 & \text{for } h < h_{\mathrm{b}}, \\ \beta & \text{for } h > h_{\mathrm{b}}, \end{cases} \tag{13}$$

and

$$\alpha_{\mathrm{q_t}}(h) = \begin{cases} 1 - \eta\epsilon & \text{for } h < h_{\mathrm{b}}, \\ \epsilon & \text{for } h > h_{\mathrm{b}}, \end{cases} \tag{14}$$

represent changes in the buoyancy flux due to latent heat release above cloud base $h_{\mathrm{b}}$. Thus, they vary for subsaturated or saturated conditions, i.e., below and above cloud base, respectively. Here, $\beta = [1 + \gamma\epsilon(\eta + 1)]/[1 + \gamma]$ considers the change in $s$ due to condensation or evaporation, with $\gamma = (L_{\mathrm{v}}/c_{\mathrm{p}}) \, \mathrm{d}q_{\mathrm{s}}/\mathrm{d}T|_{\mathrm{p_0}}$ representing the change in the saturation water vapor mixing

ratio $q_{\mathrm{s}}$ with $T$, commonly referred to the Clausius-Clapeyron relation. Typically, $\beta \approx 0.5$, $1 - \eta\epsilon \approx 0.9$, and $\epsilon \approx 0.1$, indicating that the influence of $\overline{w's'}$ and $\overline{w'q_{\mathrm{t}}'}$ on $\overline{w'b'}$ changes from below to above cloud base.

Under well-mixed conditions, contributions to the buoyancy flux that originate from the surface or the top of the boundary layer can be assumed to increase or decrease linearly within the boundary layer, reaching zero at the opposite side of the boundary layer (e.g., Bretherton and Wyant, 1997). Thus, the surface buoyancy flux can be expressed as

$$\overline{w'b'}\big|_0 (h) = \frac{g}{T_0 c_{\mathrm{p}}} \left[ \alpha_{\mathrm{s}}(h) \, \overline{w's'}\big|_0 - \alpha_{\mathrm{q_t}}(h) L_{\mathrm{v}} \, \overline{w'q_{\mathrm{t}}'}\big|_0 \right] \left( 1 - \frac{h}{h_{\mathrm{t}}} \right). \tag{15}$$

Similarly, the entrainment buoyancy flux from the layer's top yields

$$\overline{w'b'}\big|_{\mathrm{e}} (h) = \frac{g}{T_0 c_{\mathrm{p}}} \left[ -\alpha_{\mathrm{s}}(h) w_{\mathrm{e}} \Delta s + \alpha_{\mathrm{q_t}}(h) L_{\mathrm{v}} w_{\mathrm{e}} \Delta q_{\mathrm{t}} \right] \left( \frac{h}{h_{\mathrm{t}}} \right). \tag{16}$$

The effect of longwave radiation cooling is represented as

$$\overline{w'b'}\big|_{\mathrm{r}} (h) = \frac{g}{T_0 c_{\mathrm{p}}} \left[ \alpha_{\mathrm{s}}(h) \Delta F_{\mathrm{r}} \rho_0 \right] \left( \frac{h}{h_{\mathrm{t}}} \right), \tag{17}$$

which is confined to the top of the boundary layer, where longwave radiative cooling takes place primarily. Similar to Dal Gesso et al. (2014), we express the longwave radiative cooling rate as

$$\Delta F_{\mathrm{r}} = F_{\mathrm{r}}(h_{\mathrm{t}}) - F_{\mathrm{r}}(0) = \Delta F_{\mathrm{r},*} \left[ 1 - \exp\left( -\kappa_{\mathrm{r}} L \right) \right] - \lambda_{\mathrm{r}} \ln\left[ b_{\mathrm{r}}(q_{\mathrm{t},0} + \Delta q_{\mathrm{t}}) \right], \tag{18}$$

where the first term on the right-hand side describes the radiative cooling across the boundary layer, which is scaled by $L$ (e.g., Larson et al., 2007), while the second term represents the effect of free-tropospheric moisture, warming the cloud top.





**Figure 2.** A representation of (a) the vertically averaged surface buoyancy flux $\langle B \rangle|_0$, (b) entrainment buoyancy flux $\langle B \rangle|_e$, (c) longwave radiation buoyancy flux $\langle B \rangle|_r$, (d) sedimentation buoyancy flux $\langle B \rangle|_s$, (e) precipitation buoyancy flux $\langle B \rangle|_p$, and (f) sedimentation-entrainment feedback parameterization $f_A$ as a function of the liquid water path $L$ and droplet number concentration $N$. See Fig. 1 for further explanations.



$\Delta F_{r,*} = 88.2\,\mathrm{W\,m^{-2}}$, $\lambda_r = 16.5\,\mathrm{W\,m^{-2}}$, and $b_r = 1000$ have been determined from fitting $\Delta F_r$ against the free-tropospheric $q_t$ using an ensemble of stratocumulus large-eddy simulations that employed a detailed radiative-transfer code. The underlying data is shown in Fig. 9a of Chen et al. (2024b). The effective emissivity $\kappa_r$ is a function of the (effective) droplet radius at cloud top $r_l(h_t)$, and is determined by fitting the values stated in Tab. 1 of Larson et al. (2007). As all cooling is concentrated at the cloud top, (17) cannot represent the $N$-effect on the spatial distribution of longwave radiative cooling and hence entrainment,

as recently discussed by Igel (2024).

Based on Bretherton et al. (2007), we include the effect of droplet sedimentation as

$$\left.\overline{w'b'}\right|_s(h) = -gP_s(h), \tag{19}$$

where the liquid water flux by sedimentation is expressed as

$$P_s(h) = \begin{cases} 0 & \text{for } h < h_b, \\ w_t(h)q_l(h) & \text{for } h > h_b, \end{cases} \tag{20}$$

with the terminal velocity of the sedimenting droplets given by $w_t(h) = k_t r_l^2(h)$, and $k_t = 1.19 \times 10^8\,\mathrm{m^{-1}\,s^{-1}}$ (Rogers and Yau, 1989). Note that sedimenting droplets are assumed to be small enough that they do not fall below cloud base.

As we will show below, the complete evaporation of precipitation falling below cloud base (virga) can have a substantial impact on boundary layer dynamics in non-precipitating clouds at sufficiently low $N$. This effect is expressed as

$$\left.\overline{w'b'}\right|_p(h) = \frac{g}{T_0 c_p}\alpha_{q_t}(h)L_v\left[P_p(h) - P_p(0)\left(1 - \frac{h}{h_t}\right)\right]. \tag{21}$$

The precipitation liquid water flux is determined as

$$P_p(h) = \begin{cases} P_p(h_b) - [P_p(h_b) - P_p(0)]\frac{h_b - h}{h_b} & \text{for } h < h_b, \\ P_p(h_b)\left(1 - \frac{h - h_b}{h_l}\right) & \text{for } h > h_b, \end{cases} \tag{22}$$

bounded by its cloud base and surface values $P_p(h_b) = k_p(L/N)^{1.75}$ and $P_p(0) = P_p(h_b)\exp\left[-(h_b/h_p)^{1.5}\right]$, respectively, with $k_p = 2.44 \times 10^{10}\,\mathrm{kg^{-0.75}\,m^{-3.75}\,s^{-1}}$ and $h_p = 475\,\mathrm{m}$ from Wood (2007), and the geometrical cloud depth defined as

$$h_l = h_t - h_b. \tag{23}$$

Note that there are different expressions for $P_p(h_b)$ and its dependency on $N$ and $L$ discussed in the literature (e.g., Kostinski, 2008; Feingold et al., 2013), opening the potential for slight quantitative changes in the results.

The boundary-layer averaged buoyancy flux is determined from its individual components as

$$\langle B \rangle = \frac{1}{h_t}\int_0^{h_t}\overline{w'b'}(h)\,\mathrm{d}h = \frac{1}{h_t}\int_0^{h_t}\sum_i \left.\overline{w'b'}\right|_i(h)\,\mathrm{d}h = \sum_i \langle B \rangle_i, \tag{24}$$

where all contributions to $\overline{w'b'}$ are integrated analytically.





Note that $\langle B \rangle|_{\mathrm{r}}$, $\langle B \rangle|_{\mathrm{s}}$, $\langle B \rangle|_{\mathrm{p}}$ directly depend on $N$, while $\langle B \rangle|_{\mathrm{e}}$ exhibits an indirect dependency since $w_{\mathrm{e}}$ depends on $\langle B \rangle$ and hence all aforementioned direct dependencies. Moreover, to avoid the recursive dependency of $w_{\mathrm{e}}$ on $w_{\mathrm{e}}$ via $\langle B \rangle|_{\mathrm{e}}$, many parameterizations rearrange (8) to solve for $w_{\mathrm{e}}$ directly. However, such parameterizations impede the understanding of $w_{\mathrm{e}}$'s basic dependencies (cf. Stevens, 2002). Thus, we solve (8) iteratively. All contributions to $\langle B \rangle$ are presented in an $L$-$N$ phase space in Figs. 2a to e, and will be analyzed in more detail in Section 3.

### 140  2.2.3  The Efficiency Factor $A$

Based on the seminal work by Nicholls and Turton (1986) and Turton and Nicholls (1987), the entrainment efficiency can be expressed as

$$A = \frac{2}{1 - k_*} a_{\mathrm{A},1} \left[ 1 + a_{\mathrm{A},2}\, \chi_{\mathrm{A}} \left( 1 - \frac{\delta b_{\mathrm{A}}}{\Delta b} \right) f_{\mathrm{A}} \right], \tag{25}$$

where $a_{\mathrm{A},1} = 0.2$ is recommended for representing entrainment in dry boundary layers, for which the bracketed term is ne-
glected. The bracketed term represents the increase in entrainment due to the presence of liquid water at the cloud top. Yamaguchi and Randall (2012) have shown that the evaporation of liquid water mixed with free-tropospheric air cools and hence decreases the positive buoyancy of the free-tropospheric air, facilitating its subsequent entrainment. The minimum buoyancy of a mixture of boundary-layer and free-tropospheric air is achieved when all liquid water is evaporated and the mixture is exactly saturated (Stevens, 2002). Grenier and Bretherton (2001) showed that this buoyancy is

$$\delta b_{\mathrm{A}} = \frac{g}{T_0 c_{\mathrm{p}}} (\beta \Delta s - \epsilon L_{\mathrm{v}} \Delta q_{\mathrm{t}}), \tag{26}$$

with

$$\chi_{\mathrm{A}} = -\frac{q_{\mathrm{l}}(h_{\mathrm{t}})/\Delta q_{\mathrm{t}}}{1 - \frac{\gamma}{1+\gamma} \frac{\Delta s}{L_{\mathrm{v}} \Delta q_{\mathrm{t}}}} \tag{27}$$

being the necessary mass fraction of free-tropospheric air to evaporate all liquid water in this mixture. Note that other approaches to determine $\delta b_{\mathrm{A}}$ and $\chi_{\mathrm{A}}$ exist, e.g., the average over all possible mixtures (e.g., Dal Gesso et al., 2014). Further,
the parameter $a_{\mathrm{A},2} = 15$ is used in this study, as recommended by Bretherton et al. (2007), while Nicholls and Turton (1986) suggest 60.

    Lastly, Bretherton et al. (2007) introduced $f_{\mathrm{A}}$ in the bracketed term of (25) to represent the removal of liquid water from the cloud top by droplet sedimentation, thereby decreasing the potential for evaporative cooling and hence entrainment. At the same time, $f_{\mathrm{A}}$ considers that convection replenishes the liquid water. To combine these effects, Bretherton et al. (2007)
suggested

$$f_{\mathrm{A}} = \exp\left[ -a_{\mathrm{A},3} \frac{w_{\mathrm{t}}(h_{\mathrm{t}})}{w_*} \right], \tag{28}$$

where $a_{\mathrm{A},3} = 9$ is a fitting parameter and $w_{\mathrm{t}}(h_{\mathrm{t}}) = k_{\mathrm{t}} r_{\mathrm{l}}^2(h_{\mathrm{t}})$. Thus, $f_{\mathrm{A}}$ causes the bracketed term in (25) to approach 1 when sedimentation is strong ($w_{\mathrm{t}} \to \infty$) or convection weak ($w_* \to 0$).





Note that we do not include an evaporation-entrainment feedback to describe $N$-dependent evaporation at the cloud top (e.g.,
Wang et al., 2003; Igel, 2024). While no parameterization for this effect exists, it is reasonable to assume that it should behave
similarly to the sedimentation-entrainment feedback described by (28). Thus, increasing the magnitude of $a_{A,3}$ in (28) can be
seen as a means to estimate the effect of evaporation-entrainment, and is done in Section 3.3.

### 2.3 Determining Cloud-Water Adjustments

#### 2.3.1 Relations to $L$ and $N$

To understand cloud-water adjustments due to entrainment, we would like to express $w_e$ as a function of $L$ and $N$. We start by
determining $q_l$ as a function of the distance to the cloud base $h_b$, i.e.,

$$q_l(h) = \begin{cases} 0 & \text{for } h < h_b, \\ \Gamma_l(h - h_b) & \text{for } h > h_b, \end{cases} \tag{29}$$

where $\Gamma_l$ describes the increase of liquid water mixing ratio with height (e.g., Albrecht et al., 1990). Thus,

$$L = \int_0^{h_t} \rho_0 q_l(h)\, \mathrm{d}h = \frac{1}{2}\rho_0 \Gamma_l h_l^2, \tag{30}$$

using (23) and the reference air density $\rho_0$. This definition allows us to express $h_l$ as

$$h_l = \left( \frac{2L}{\rho_0 \Gamma_l} \right)^{1/2}, \tag{31}$$

and the corresponding cloud-top value of $q_l$ as

$$q_l(h_t) = \Gamma_l h_l = \left( \frac{2\Gamma_l L}{\rho_0} \right)^{1/2}. \tag{32}$$

The droplet radius changes with $h$ as

$$r_l(h) = \left[ \frac{q_l(h)\rho_0}{\frac{4}{3}\pi \rho_l N} \right]^{1/3}, \tag{33}$$

and the corresponding cloud-top value is

$$r_l(h_t) = \left( \frac{9\Gamma_l L \rho_0}{8\pi^2 \rho_l^2 N^2} \right)^{1/6}. \tag{34}$$

By prescribing $L$ and $N$, as well as the parameters $h_t$, $T_0$, and $\rho_0$, (31) to (34) can be used to express $\Delta b$, $\langle B \rangle$, $A$, and hence
$w_e$ as a function of $L$ and $N$. Thus, $s$ and $q_t$ are not required to determine $\Delta b$, $\langle B \rangle$, $A$, and $w_e$, but are assumed to adjust such
that a prescribed value of $L$ is obtained.





### 2.3.2 Base Assumptions

The large-eddy simulations in our companion paper (Chen et al., 2024a) show that a positive perturbation of $N$, $\delta N > 0$, results in an increase in $w_e$ in response to an aerosol perturbation (see their Fig. 4a). After sufficient time (18 h), this increase in $w_e$ is depleted, resulting in negligible differences in $w_e$ among the perturbed and unperturbed simulations. This decrease in $w_e$ is enabled by a commensurate decrease in $L$ in the perturbed simulations, $\delta L < 0$, resulting in increasingly stronger negative $m$ with time (see their Fig. 2c). Note that other parameters assumed as constant in our framework (e.g., $h_t$) change insignificantly among the perturbed and unperturbed simulations (see their Fig. 4b). We approximate this late-stage behavior by

$$w_e(N, L) = w_e(N + \delta N, L + \delta L). \tag{35}$$

Using the mixed-layer model entrainment parameterization $w_e$ outlined above, (35) is solved iteratively for $\delta L$ using prescribed values of $L$, $N$, and $\delta N$, while keeping all other parameters constant. Then, cloud-water adjustments are quantified as $m \approx [\ln(L + \delta L) - \ln(L)]/[\ln(N + \delta N) - \ln(N)]$.

Note that (35) describes a condition that is assumed to be valid *in addition* to other changes affecting $L$ and $N$. Since (35) is only valid after sufficient time (18 h) (Chen et al., 2024a), stratocumulus that exhibit faster changes in $L$ and $N$ should not be assessed using (35). This might be the case for stratocumulus that are far from their steady state $L$ (Hoffmann et al., 2020; Glassmeier et al., 2021; Hoffmann et al., 2024b). On the other hand, slower processes that affect $h_t$ or $L$ on longer timescales do not influence our assessment (Schubert et al., 1979; Stevens, 2006; Bretherton et al., 2010).

Moreover, it is important to reiterate that the $m$ determined from (35) is only valid for stratocumulus-topped boundary layers that are driven by the interplay of entrainment and longwave radiation (Hoffmann et al., 2020), with a negligible contribution of surface precipitation. Surface precipitation would constitute a loss term for $L$ that is not considered in (35). Thus, $m$ should only be interpreted for parts of the $L$-$N$ phase space where losses by surface precipitation are negligible. Positive $m$, traditionally associated with precipitation suppression (e.g., Albrecht, 1989), are not part of our solution.

### 2.4 Parameters

The default stratocumulus-topped boundary layer analyzed here is based on the large-eddy intercomparison case by Ackerman et al. (2009), derived from the second research flight of the DYCOMS-II campaign that focused on subtropical stratocumulus off the coast of California (Stevens et al., 2003). Here, we use their surface fluxes $\overline{w'q_t'}|_0 = 93\,\mathrm{W\,m^{-2}}/(L_v \rho_0)$ and $\overline{w's'}|_0 = 16\,\mathrm{W\,m^{-2}}/\rho_0 + L_v \overline{w'q_t'}|_0 = 109\,\mathrm{W\,m^{-2}}/\rho_0$, jumps $\Delta q_t = -4.45\,\mathrm{g\,kg^{-1}}$ and $\Delta s/c_p = 6.7\,\mathrm{K} + \Delta q_t L_v/c_p \approx -3.3\,\mathrm{K}$, and boundary layer depth $h_t = 795\,\mathrm{m}$, unless otherwise noted. To estimate the free-tropospheric humidity for determining the effect of longwave radiative cooling (18), we prescribe a $q_{t,0} = 9.45\,\mathrm{g\,kg^{-1}}$ based on the initial values of Ackerman et al. (2009). Note that this $q_{t,0}$ is not used anywhere else in our framework. Other reference parameters are $T_0 = 288.3\,\mathrm{K}$, $\rho_0 = 1.21\,\mathrm{kg\,m^{-3}}$, and $\Gamma_l = 2\,\mathrm{g\,kg^{-1}\,km^{-1}}$.





## 3 Results

Most plots of this study show the $L$-$N$ phase space, overlayed by different lines and patterns. The stippling marks potentially decoupled boundary layers, where the buoyancy flux is too weak to ensure a well-mixed boundary layer. These regions have been determined using the approach by Turton and Nicholls (1987). Reasons for the decoupling will be discussed more deeply

when addressing $\langle B \rangle$. As decoupled boundary layers violate many assumptions reasonable for well-mixed boundary layers, this part of the phase space should not be assessed. Moreover, the striped part of the phase space marks regions where the cloud base is at or below the surface ($h_\mathrm{b} < 0$), and results should be disregarded. The dashed line marks the *surface* precipitation rate of $P_\mathrm{p}(0) = 0.1\,\overline{w'q_\mathrm{t}'}|_0\,\rho_0 \approx 0.3\,\mathrm{mm\,d}^{-1}$. By determining this value based on $\overline{w'q_\mathrm{t}'}|_0$, we identify regions of the $L$-$N$ phase space in which the $L$ budget is substantially affected by precipitation losses (to the left of the dashed line) and the region where

precipitation losses are negligible, i.e., the non-precipitating clouds that are the main focus of this study (to the right of the dashed line). Note that the precipitation rate to discriminate these regions is comparable to the *cloud base* precipitation rate of $0.5\,\mathrm{mm\,d}^{-1}$ suggested by Wood (2012).

### 3.1 Entrainment at High $N$

Figure 1a shows $w_\mathrm{e}$ as a function of $L$ and $N$. First, we address $w_\mathrm{e}$ in the non-precipitating part of the phase space between

$N = 300$ and $1000\,\mathrm{cm}^{-3}$, i.e., sufficiently far away from the precipitating phase space. Here, we see a strong increase in $w_\mathrm{e}$ with $L$, accompanied by a weak dependency on $N$. The strong increase in $w_\mathrm{e}$ with $L$ can be attributed to a decrease in $\Delta b$ (Fig. 1c) and an increase in $A$ (Fig. 1b). The decrease in $\Delta b$ is due to the stronger latent heat release at higher $L$, which decreases the temperature difference relative to the warmer free troposphere, as indicated by (11), enabling stronger entrainment. Note that $\Delta b$ does not depend on $N$. The increase in $A$ with $L$ is primarily due to the larger amount of liquid water at the cloud

top, requiring a much larger fraction of free-tropospheric air to be mixed with cloudy air to evaporate it according to (27), thus allowing more free-tropospheric air to be entrained into the boundary layer. For a given $N$, this effect is somewhat offset by the sedimentation-entrainment feedback (28) (Fig. 2f), where an increase in droplet size with $L$ increases the removal of liquid water from the cloud top by sedimentation, which decreases entrainment. Interestingly, the sedimentation-entrainment feedback decreases for sufficiently high $L$. This is because the sedimentation-entrainment feedback is proportional to $w_\mathrm{t}/w_*$,

and the increase in $w_*$ for $L > 300\,\mathrm{g\,m}^{-2}$ (cf. $\langle B \rangle \sim w_*^3$ in Fig. 1d) outpaces the simultaneous increase in $w_\mathrm{t}$. The slight increase of $A$ with $N$ is due to the weakening of the sedimentation-entrainment feedback for higher $N$, where decreasing droplet sedimentation from the cloud top enables stronger entrainment.

    While $\langle B \rangle$ also affects $w_\mathrm{e}$, its variability between $N = 300$ and $1000\,\mathrm{cm}^{-3}$ is much weaker than the variability in $\Delta b$ and $A$. Nonetheless, for $L < 30\,\mathrm{g\,m}^{-2}$, $\langle B \rangle$ exhibits a strong increase with $L$ due to increasing longwave radiative cooling accelerating

boundary layer dynamics, as shown by $\langle B \rangle|_\mathrm{r}$ (Fig. 2c). Note that the increase in longwave radiative cooling quickly saturates for $L > 30\,\mathrm{g\,m}^{-2}$ (Garrett et al., 2002). However, $\langle B \rangle|_\mathrm{r}$ decreases slightly after reaching a maximum at around $L = 30\,\mathrm{g\,m}^{-2}$. The coefficient (13) in (17) indicates that $\overline{w'b'}|_\mathrm{r}$ is smaller in saturated layers than in subsaturated ones. Because $h_\mathrm{t}$ is kept constant, the partitioning of the boundary layer into saturated and subsaturated layers is determined by $L$, with larger $L$ resulting in





deeper saturated layers. Accordingly, the vertical average over $\overline{w'b'}|_\mathrm{r}$, $\langle B \rangle|_\mathrm{r}$, will decrease as $L$ increases. In addition to the

variability with $L$, $\langle B \rangle|_\mathrm{r}$ increases substantially with $N$ for $L < 30\,\mathrm{g\,m}^{-2}$. This is due to the dependency of the longwave radiative effective emissivity ($\kappa_\mathrm{r}$) on droplet size considered in (18), which has a substantial impact as long as longwave radiative cooling is not saturated.

The (constant) surface fluxes do not substantially affect $\langle B \rangle$, although $\langle B \rangle|_0$ increases slightly for $L > 300\,\mathrm{g\,m}^{-2}$ (Fig. 2a). As above, this increase is caused by the $L$-dependent partitioning of the boundary layer in subsaturated and saturated parts,

where $\overline{w'b'}|_0$ tends to increase in a saturated environment due to (13) and (14) in (15).

The impact of entrainment on $\langle B \rangle$ varies substantially with $L$ (Fig. 2b). For $L < 300\,\mathrm{g\,m}^{-2}$, $\langle B \rangle|_\mathrm{e}$ is negative, as entrainment introduces warmer air that stabilizes the dynamics of the boundary layer from its top. This effect can be so strong that it decouples the cloud layer from the sub-cloud layer when other buoyancy sources are insufficient (stippled part of the non-precipitating phase space). For $L > 300\,\mathrm{g\,m}^{-2}$, however, $\langle B \rangle|_\mathrm{e}$ increases substantially. This change can be interpreted as a

form of cloud-top entrainment instability (CTEI) that intensifies the dynamics of the boundary layer and hence entrainment as a response to entrainment (e.g., Lilly, 1968; Randall, 1980). Here, the reason for this is, again, the $L$-dependent partitioning of the boundary layer into subsaturated and saturated parts. In the subsaturated layer, $\overline{w'b'}|_\mathrm{e} < 0$ according to (16) with (13) and (14), while $\overline{w'b'}|_\mathrm{e} > 0$ in the saturated part. Thus, averaging over a boundary layer with sufficiently large $L$ will result in a $\langle B \rangle|_\mathrm{e} > 0$. Note that this effect has to be distinguished from the more traditional depiction of CTEI, which is included in $A$ via

(26) and (27) (Stevens, 2002).

## 3.2 Entrainment at Lower $N$

Now, we address the part of the $L$-$N$ phase space below $N = 300\,\mathrm{cm}^{-3}$ but above the assumed critical surface precipitation rate (right of the dashed line), where a much stronger dependency of $w_\mathrm{e}$, $\langle B \rangle$, and $A$ on $N$ is recognized (Figs. 1a, b, and d), while $\Delta b$ is independent of $N$ (Fig. 1c).

The dependency of $w_\mathrm{e}$ on $N$ is primarily due to $\langle B \rangle$, which decreases substantially due to precipitation evaporating below cloud base. The associated cooling stabilizes boundary layer dynamics, i.e., causes a negative $\langle B \rangle|_\mathrm{p}$ (Fig. 2e), which decreases entrainment. The magnitude of $\langle B \rangle|_\mathrm{p}$ increases with $L$ and decreases with $N$, as expected from the corresponding effect on the mean droplet size and hence the precipitation flux (22). Interestingly, $\langle B \rangle|_\mathrm{p}$ vanishes for $L > 300\,\mathrm{g\,m}^{-2}$. This is due to the diminishing sub-cloud layer depth necessary for precipitation to evaporate. Similarly, sedimentation also decreases $\langle B \rangle$, with

the magnitude of $\langle B \rangle|_\mathrm{s}$ increasing with $L$ and decreasing with $N$ (Fig. 2d), while the influence is much smaller than that of precipitation.

The dependence of $A$ on $L$ for $N$ slightly below $N = 300\,\mathrm{cm}^{-3}$ resembles that for higher $N$ (Fig. 1b), overlayed with a stronger sedimentation-entrainment feedback increasing the $N$ dependency (Fig. 2f). This behavior is caused by a stronger sedimentation at lower $N$, and enhanced by the strong decay in boundary layer dynamics (cf. $\langle B \rangle \sim w_*^3$ in Fig. 1d), limiting

the replenishment of liquid water at the cloud top due to the aforementioned stabilizing effect of evaporating precipitation.

In the precipitating part of the phase space (left of the dashed line), the stabilizing effect of evaporating precipitation can become so strong that $\langle B \rangle$ becomes negative (Figs. 1d and 2e), which decouples the cloud layer from the sub-cloud layer (stip-





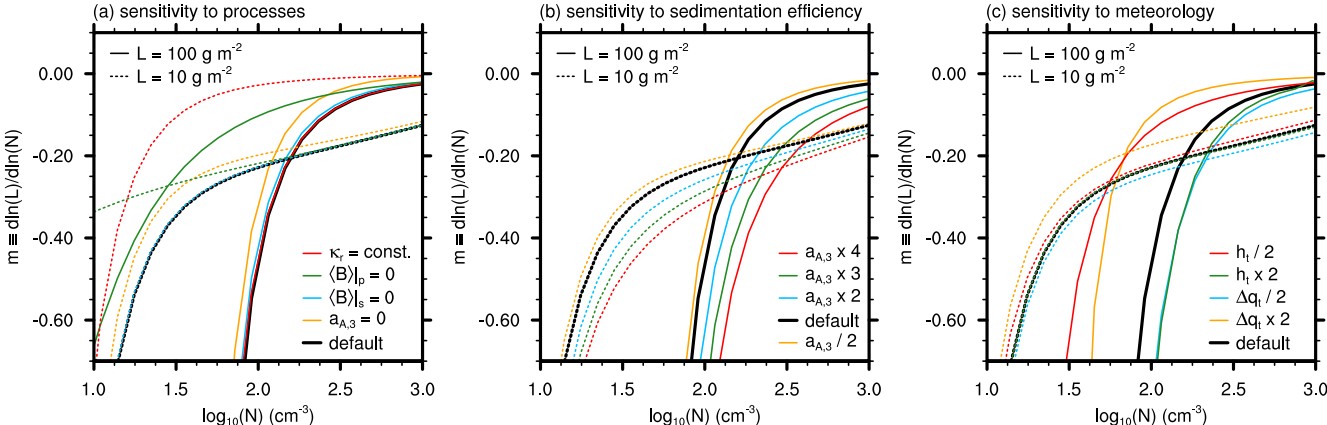

**Figure 3.** Panels (a) to (c) show $m$ as a function of $N$ for $L = 10\,\mathrm{g\,m^{-2}}$ (dashed lines) and $100\,\mathrm{g\,m^{-2}}$ (continuous lines), with a thick black line indicating the default case. Colored lines highlight the sensitivity of $m$ to (a) processes, (b) sedimentation efficiency, and (c) meteorology.

pled part of the relevant phase space). However, it is important to recognize that precipitation is not limited to the precipitating part of the phase space, but also occurs in the non-precipitating part at sufficiently low $N$. Here, only weak precipitation is
produced by the cloud and largely evaporated below cloud base, commonly referred to as virga.

### 3.3 Cloud-Water Adjustments and its Sensitivities

Figure 3 shows $m \equiv \mathrm{d\ln}(L)/\mathrm{d\ln}(N)$ as a function of $N$ for $L = 10$ and $100\,\mathrm{g\,m^{-2}}$ (dashed and continuous lines, respectively), representing cases with unsaturated and saturated longwave radiative cooling. The case analyzed above (the default) is indicated by black lines, while altered setups are indicated by colored lines. For clarity, we will focus our discussion on how $m$ changes
between $N = 100$ and $1000\,\mathrm{cm^{-3}}$, typical values that circumscribe non-precipitating stratocumulus. For the default case, $m$ weakens from $-0.48$ at $N = 100\,\mathrm{cm^{-3}}$ to $-0.03$ at $N = 1000\,\mathrm{cm^{-3}}$ for $L = 100\,\mathrm{g\,m^{-2}}$, while $m$ changes from $-0.23$ to $-0.13$ for $L = 10\,\mathrm{g\,m^{-2}}$ (black lines in Fig. 3). In both cases, we see that $m$ is not constant for non-precipitating stratocumulus, which is in agreement with recent large-eddy simulations presented in our companion paper (Chen et al., 2024a) (see their Figs. 1b and d).
Figure 3a shows how $m$ depends on the representation of various processes considered in the estimate of $w_e$. To remove the dependence of the emission of longwave radiation on $N$, a constant effective emissivity $\kappa_r = 85\,\mathrm{m^2\,kg^{-1}}$ instead of the radius-dependent $\kappa_r$ used above is applied (red lines). For $L = 100\,\mathrm{g\,m^{-2}}$, $m$ is not substantially affected by this change, which is expected since the emission of longwave radiation is saturated. For $L = 10\,\mathrm{g\,m^{-2}}$, however, $m$ weakens substantially from $-0.23$ to negligible $-0.03$ at $N = 100\,\mathrm{cm^{-3}}$ and vanishes completely for higher $N$. This indicates that the dependency of
longwave radiation on $N$ is the major driver for cloud water adjustments in non-precipitating clouds at low $L$. This picture reverses when the influence of precipitation on $\langle B \rangle$ is neglected by setting $\langle B \rangle|_p = 0$ (green lines). For $N = 100\,\mathrm{cm^{-3}}$, the absence of precipitation weakens $m$ from $-0.48$ to $-0.10$ for $L = 100\,\mathrm{g\,m^{-2}}$, while it changes negligibly from $-0.23$ to $-0.22$



for $L = 10\,\mathrm{g\,m^{-2}}$. At $N = 1000\,\mathrm{cm^{-3}}$, inhibiting precipitation does not change $m$ for any $L$. Thus, evaporating precipitation is the main driver for $m$ for $N = 100\,\mathrm{cm^{-3}}$ and high $L$, while it does not affect $m$ at low $L$ or high $N$.

We disregard the sedimentation-entrainment effect by setting $a_{\mathrm{A},3} = 0$ in (28) (orange lines). For $L = 100\,\mathrm{g\,m^{-2}}$, the absence of sedimentation-entrainment weakens $m$ from $-0.48$ to $-0.32$ at $N = 100\,\mathrm{cm^{-3}}$, indicating a substantial but smaller impact than evaporating precipitation. For higher $N$ and lower $L$, sedimentation becomes weaker, as does the influence of the sedimentation-entrainment feedback. However, the sedimentation-entrainment feedback is still responsible for the bulk of cloud-water adjustments for $L = 100\,\mathrm{g\,m^{-2}}$ at $N = 1000\,\mathrm{cm^{-3}}$, where its absence weakens $m$ from $-0.03$ to $-0.01$. For
$L = 10\,\mathrm{g\,m^{-2}}$, on the other hand, longwave radiation exceeds the impact of the sedimentation-entrainment feedback for all analyzed $N$, with $m$ weakening only from $-0.23$ to $-0.20$ at $N = 100\,\mathrm{cm^{-3}}$ and $-0.14$ to $-0.12$ at $N = 1000\,\mathrm{cm^{-3}}$ in the absence of the sedimentation-entrainment feedbacks. Note that neglecting sedimentation effects on $\langle B \rangle$ by setting $\langle B \rangle|_{\mathrm{s}} = 0$ (blue lines) affects $m$ negligibly for all $L$ between $N = 100$ and $1000\,\mathrm{cm^{-3}}$.

High resolution modeling (direct numerical simulations) by de Lozar and Mellado (2017) indicate that the sedimentation-
entrainment feedback can be about three times stronger than the estimate by Bretherton et al. (2007) based on large-eddy simulations. To analyze this, we varied $a_{\mathrm{A},3}$ from half to four times its default value (Fig. 3b). As expected, the effect of sedimentation-entrainment is proportional to $a_{\mathrm{A},3}$ and mostly visible for high $L$. For $L = 100\,\mathrm{g\,m^{-2}}$ and $a_{\mathrm{A},3} \times 3$ (continuous green line), we see a substantial strengthening of $m$ from $-0.48$ to $-0.80$ at $N = 100\,\mathrm{cm^{-3}}$ and $-0.03$ to $-0.06$ at $N = 1000\,\mathrm{cm^{-3}}$, making the sedimentation-entrainment feedback comparable to the effect of evaporating precipitation at low $N$
(Fig. 3a), while also allowing it to persist for higher $N$.

In Fig. 3c, we analyze the influence of two external (meteorological) parameters on $m$: boundary layer depth and free-tropospheric humidity, which are often considered to affect cloud-water adjustments (e.g., Possner et al., 2020; Glassmeier et al., 2021). To address this, we vary $h_{\mathrm{t}}$ (red and green lines) and $\Delta q_{\mathrm{t}}$ (blue and orange lines) by halving or doubling their default values. For $L = 100\,\mathrm{g\,m^{-2}}$, we find that $m$ becomes more negative with $h_{\mathrm{t}}$, in agreement with Possner et al. (2020)
(red to black to green continuous lines). For $L = 10\,\mathrm{g\,m^{-2}}$, however, there is almost no effect of $h_{\mathrm{t}}$ (red to black to green dashed lines). This $L$-dependence indicates that the main reason for the $h_{\mathrm{t}}$-dependence of $m$ is the increased potential for evaporation of precipitation due to a deeper sub-cloud layer at higher $h_{\mathrm{t}}$. Interestingly, we find that a drier free troposphere ($\Delta q_{\mathrm{t}} \times 2$, orange lines) weakens cloud-water adjustments, while a moister free troposphere ($\Delta q_{\mathrm{t}}/2$, blue lines) increases them, which is in contrast to Glassmeier et al. (2021), but in agreement with recent large-eddy simulations by Chun et al. (2023) and
Chen et al. (2024a). The reason for this behavior is twofold. First, increased moisture in the free-troposphere reduces longwave radiative cooling at the cloud top, as considered in (18). Accordingly, the relative influence of $L$ on longwave radiative cooling decreases, and a much stronger decrease in $L$ is necessary to offset an increase in $w_{\mathrm{e}}$ to fulfill (35). At the same time, a moister free troposphere is more positively buoyant, which increases $\Delta b$. As above, a much stronger decrease in $L$ is necessary to offset an increase in $w_{\mathrm{e}}$ in this situation.



## 4 Summary and Conclusions

Cloud-water adjustments strongly modify the ability of clouds to reflect shortwave radiation, and hence their role in Earth's radiation budget (e.g., Stevens and Feingold, 2009). However, the magnitude and even the sign of cloud-water adjustments is insufficiently understood (e.g., Boucher et al., 2013; Forster et al., 2021). This is especially true for non-precipitating stratocumulus, in which various processes tend to increase the mixing of the cloud with the free-troposphere (entrainment) in response to an increase in aerosol and hence droplet concentration $N$, resulting in a decrease in the vertically integrated cloud water content $L$ due to evaporation.

This study was built upon the assumption that an increase in entrainment rate $w_e$ due to an increase in $N$ is exactly offset by a commensurate decrease in $L$, resulting in the same $w_e$ irrespective of $N$. This idea is based on large-eddy simulations presented in our companion paper (Chen et al., 2024a) (see their Fig. 4a), and can be considered a corollary of the $w_e$-$L$ feedback mechanism suggested by Zhu et al. (2005) (see their Fig. 7). This assumption, combined with a $w_e$ parameterization developed for stratocumulus-topped boundary layers (Nicholls and Turton, 1986; Turton and Nicholls, 1987), enabled us to determine $m \equiv \mathrm{d}\ln(L)/\mathrm{d}\ln(N)$, the common metric to quantify cloud-water adjustments, in an almost completely analytical way. Note that the $m$ derived in this study is only valid for stratocumulus with negligible surface precipitation, and the effect of precipitation suppression on $m$ (e.g., Albrecht, 1989) was not considered. Moreover, real clouds might not fully offset the increase of $w_e$ due to an increase in $N$, with commensurate impacts on $L$ and $m$. Thus, this study does not aim for a full quantitative understanding of cloud-water adjustments, but to untangle the effects of various processes comprising cloud-water adjustments of non-precipitating stratocumulus.

We showed that three processes are mainly responsible for negative cloud water adjustments in non-precipitating stratocumulus: First, we showed that the full evaporation of precipitation below cloud base, commonly referred to as virga, is a major source for negative cloud-water adjustments (Caldwell et al., 2005; Wood, 2007; Sandu et al., 2008; Uchida et al., 2010). Although virga affects only a limited range of non-precipitating stratocumulus at low $N$, it is able to stabilize boundary layer dynamics and hence to decrease $w_e$. Thus, an increase in $N$ strengthens boundary layer dynamics and hence $w_e$, with commensurate negative impacts on $L$. Second, $w_e$ decreases when sedimentation removes liquid water from the cloud top (Bretherton et al., 2007), which limits the necessary preconditioning of to-be-entrained air by the evaporation of cloud water (Yamaguchi and Randall, 2012). Reducing the sedimentation of liquid water by an increase in $N$, $L$ decreases due to an increase in $w_e$. This so-called sedimentation-entrainment feedback is found to affect clouds with sufficiently high $L$, but is exceeded by the aforementioned effect of virga at low $N$. However, the sedimentation-entrainment feedback becomes the primary source of negative cloud water adjustments at high $N$. Note that the sedimentation-entrainment feedback could be stronger according to modeling by de Lozar and Mellado (2017). Third, we showed that the droplet-size dependence of longwave radiation causes negative cloud-water adjustments, but only when longwave radiative cooling is not saturated, i.e., for clouds with $L < 30\,\mathrm{g\,m^{-2}}$ (e.g., Garrett et al., 2002; Petters et al., 2012).

Previously, it has been suggested to describe cloud water adjustments in non-precipitating stratocumulus by a constant $m < 0$ (Gryspeerdt et al., 2019; Glassmeier et al., 2021). Our study showed, however, that these negative cloud water adjustments tend



to be relatively strong at small $N$ ($m = -0.48$ at $N = 100\,\text{cm}^{-3}$) and weaken toward larger $N$ ($m = -0.03$ at $N = 1000\,\text{cm}^{-3}$)

due to the cessation of precipitation and sedimentation effects. This weakening of $m$ toward larger $N$ is found to be slower for $L < 30\,\text{g}\,\text{m}^{-2}$, where unsaturated longwave radiation has to be considered. A similar weakening of negative cloud-water adjustments is shown in the large-eddy simulations of our companion paper (Chen et al., 2024a) (see their Figs. 1b and d).

Moreover, our study showed that deeper boundary layers exhibit stronger negative cloud-water adjustments in agreement with Possner et al. (2020), which we explained by the increased potential for the full evaporation of precipitation below cloud

base, i.e., a higher potential for virga. Further, we showed that increasing free-tropospheric humidity strengthens negative cloud-water adjustments, in contrast to modeling by Glassmeier et al. (2021), but in agreement with Chun et al. (2023) and our companion paper (Chen et al., 2024a). Because $w_\text{e}$ weakens as the free-tropospheric humidities increases, stronger (relative) reductions in $L$ were necessary to offset the increase of $w_\text{e}$ following an increase in $N$.

Finally, we would like to speculate on the timescales associated with the negative cloud-water adjustments laid out in this

study. Our results suggest that these adjustments are required to accelerate boundary layer dynamics when the stabilizing effect of virga is suppressed by an increase in $N$. At higher $N$, however, negative cloud-water adjustments are determined by the increasing supply of preconditioned free-tropospheric air (Yamaguchi and Randall, 2012; Bretherton et al., 2007). One might argue that the timescale associated with this process is shorter, as it does not require the acceleration of boundary layer dynamics (cf. Feingold et al., 2015). (The vertically averaged buoyancy flux $\langle B \rangle$ in Fig. 1d exhibits much stronger gradient

toward higher $N$ at low $N$ than at high $N$.) While our work cannot determine timescales associated with the changes in boundary layer dynamics, it does suggest that the timescale for cloud-water adjustments should decrease toward higher $N$.

All in all, this study showed that even comparably simple models, as the one used here, can be applied to increase our fundamental understanding of aerosol-cloud interactions. In fact, the simplicity of the applied model allowed us to directly link cause and effect of cloud-water adjustments, which can be difficult in more complex models, such as global circulation

or even large-eddy simulation models due to confounding factors (Mülmenstädt et al., 2024). Thus, the assessment of aerosol-cloud interactions should balance the use of complex and simple approaches by substantiating quantitative understanding with qualitative insight.

*Data availability.* The data to reproduce Figs. 1 to 3 is archived in a repository (Hoffmann et al., 2024a).

*Author contributions.* FH, YC, and GF conceived the study. FH carried out the study and has written the initial manuscript. YC and GF

revised the manuscript.

*Competing interests.* GF is a co-editor of ACP. Other than that, the authors declare that they have no competing interests.



*Acknowledgements.* FH appreciates support from the Emmy Noether program of the German Research Foundation (DFG) under grant HO 6588/1-1. GF acknowledges funding from NOAA's ERB program (NOAA CPO Climate and CI 03-01-07-001).



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
