# Peer review of "On the Processes Determining the Slope of Cloud-Water Adjustments in Weakly and Non-Precipitating Stratocumulus"

_EGUsphere, 2024_

## Author Comment (AC1)

Response to Reviewer 1

The paper addresses how the cloud droplet concentration (N) affects the liquid water path (LWP) in terms of a factor m = d ln (LWP)/d ln (N). The paper contains some interesting ideas, and could be a nice addition to existing MLM studies. However, a central question is whether the MLM framework, and in particular the set up of the experiments in terms of construction of the buoyancy flux from the fluxes of energy and water (do they give a steady-state?), and the boundary conditions (constant surface fluxes?) justifies a generalization of the results. The results may depend strongly on the assumptions made in the model, like a zero entrainment response to simultaneous changes in N and the liquid water path (see the sentence "The conclusions drawn in this study are built upon the assumption that an increase in entrainment rate (w_e) due to an increase in N is exactly offset by a commensurate decrease in LWP, resulting in the same w_e irrespective of N".) I recommend to pay more attention to the validity of this assumption, in particular for some extreme values of N and LWP used in the study. Another concern is that the model framework builds on existing mixed layer model studies, but for the reader not familiar with this modeling technique it may be difficult to follow.

We thank the reviewer for the valuable comments that helped to clarify various aspects of our manuscript. Our answers are stated in green font, while excerpts from the (revised) manuscript are printed in blue font with italic characters. All references not explicitly stated in this letter can be obtained from the revised manuscript.

Main:

The conclusions of this study are  "... built upon the assumption that an increase in entrainment rate (w_e) due to an increase in N is exactly offset by a commensurate decrease in L, resulting in the same we irrespective of N".

Please note that we will refer to this assumption as assumption (35) in the following.

The setup is somewhat unclear, and there may be inconsistencies, though I may be mistaken. For example, the surface fluxes are treated as constant, which seems to contradict the systematic changes in LWP. Since the boundary layer height is kept constant, the changes in LWP must stem from changes in humidity or temperature within the boundary layer, or both. I suspect that the thermodynamic profiles in the boundary layer are altered, as inferred from the dependence of the buoyancy jump (Delta b, line 183) on the settings. However, if the thermodynamic profiles in the boundary layer are modified, this would lead to changes in surface fluxes (see Bretherton and Wyant, 1997). Notably, most MLM studies cited in this paper (e.g., Wood 2007, Dal Gesso, Jones et al.) use surface boundary conditions dependent on wind speed and the difference between surface conditions and the air just above it.

First, we would like to clarify that did not use a full mixed layer model, which would be necessary to address some of the aforementioned effect. We only used the entrainment rate parameterization frequently used in mixed layer models, as stated in the abstract: *"In this study, we utilize a simple entrainment parameterization used in mixed-layer models to determine entrainment- mediated cloud-water adjustments in non-precipitating stratocumulus."* In 2.3, we further clarify that *"Using the mixed-layer model entrainment parameterization $w_e$ outlined above, (35) is solved iteratively for $\delta L$ using prescribed values of L, N , and $\delta N$ , while keeping all*

*other parameters constant."* We chose this this simplified framework to gain a better qualitative understanding of cloud-water adjustment, which in weakly and non-precipitating stratocumulus is believed to be mainly a result of changes in entrainment (e.g., Hoffmann et al. 2020).

To only rely on the entrainment parameterization, many parameters had to be assumed constant to provide a path forward to better understand the underlying physics. In particular, we do not aim for a quantitative understanding, for which more complex models, such as large-eddy simulations (LESs), are indeed necessary. In fact, the LESs of our companion paper (Chen et al. 2024) consider all the processes mentioned by the reviewer (a non-constant boundary layer depth, interactive surface fluxes), except shortwave radiation which we have studied in LESs in a previous study (Chen et al. 2024b).

One clear result of our companion paper is that the entrainment rate increases after an initial perturbation in the cloud droplet concentration N, but approaches the unperturbed value after approximately 18 h. This is shown in Fig. 4a of our companion paper (Chen et al. 2024a), which depicts a time series of the entrainment rate for different perturbations in the cloud droplet concentration. Figure 2a of our companion paper (Chen et al. 2024a) indicates that this return of the entrainment rate to its unperturbed state is mainly due to a decrease in the liquid water path. Thus, we believe that the idealization that the entrainment rate of a perturbed case assumes its unperturbed value due to a decrease in liquid water is sufficient to understand this system better.

Overall, this idealization (35) can be considered a corollary of the feedback mechanism between entrainment rate and liquid water path originally suggested by Zhu et al. (2005) (see their Fig. 7), which has been also highlighted by Wood (2012) as a major mechanism responsible for the observed stability of stratocumulus (see their Fig. 26). Although there is certainly more nuance, e.g., in the variability of surface fluxes or the boundary layer depth in response to an aerosol perturbation, our simple model is sufficient to capture the main response of non-precipitating stratocumulus to perturbations in the aerosol concentration. Thus, idealizing the underlying system to focus on the main drivers of cloud-water-adjustments in weakly and non-precipitating stratocumulus can yield reasonable and – above all – new insights. We amended our last paragraph accordingly: *"All in all, this study showed that even comparably simple models, such as the one used here, can be applied to increase our fundamental understanding of aerosol-cloud interactions. In fact, the simplicity of the applied model allowed us to directly link cause and effect of cloud-water adjustments, which can be difficult in more complex models such as global circulation or even large-eddy simulation models due to confounding factors (Mülmenstädt et al., 2024). That being said, the nuance provided by these models should not be disregarded as they help to quantify the effects that have been neglected here (e.g., interactive surface fluxes, changes in boundary layer depth, or the diurnal cycle of solar radiation). Thus, we advocate that the assessment of aerosol-cloud interactions should balance the use of complex and simple approaches by substantiating quantitative understanding with qualitative insight."*

Moreover, we added the following statement to Section 2.3.2 on the model's base assumptions: *"Additionally, (35) does not consider adjustments in surface fluxes (Bretherton and Wyant, 1997), as well as the impact of solar radiation (Chen et al., 2024b), for which more complex models would be required. Nonetheless, the large-eddy simulations presented in our companion*

*paper (Chen et al., 2024a) indicate that sufficient physics are captured in (35) to yield reasonable insights, as we will detail below."*

A particularly confusing sentence appears on line 232: "The decrease in Delta b is due to the stronger latent heat release at higher L, which decreases the temperature difference relative to the warmer free troposphere, as indicated by (11), enabling stronger entrainment." Here the factor Delta b (the vertical static energy jump across the inversion) depends on the inversion jumps of temperature and humidity, yet the sentence suggests that the boundary layer is warming (i.e., "decreasing the temperature difference", as stated). If this is the case, shouldn't the surface fluxes also change?

As stated above, we do not use a full mixed layer model, which would be necessary to address this effect. We only use an entrainment rate parameterization frequently used in mixed layer models, as stated in the abstract: *"In this study, we utilize a simple entrainment parameterization used in mixed-layer models to determine entrainment- mediated cloud-water adjustments in non-precipitating stratocumulus."* In 2.3, we further clarify that *"Using the mixed-layer model entrainment parameterization $w_e$ outlined above, (35) is solved iteratively for $\delta L$ using prescribed values of L, N , and $\delta N$ , while keeping all other parameters constant."* As already outlined above, we believe that this simplified framework is necessary to gain a better qualitative understanding of cloud-water adjustment, while more nuanced answers would require more sophisticated approaches, e.g., a full mixed layer model or large-eddy simulations, creating a commensurately more difficult path forward to understand aerosol-cloud interactions.

In Section 2.3.2 it is unclear whether an equilibrium state is assumed? In any case the boundary layer depth $h_t$ is assumed to be constant, and this leads to Eq 35, $w_e(N,LWP)=w_e(N+dN,L+dL)$, with $w_e$ being the entrainment velocity. The question arises whether Eq. 35 holds for large deviations dN and dL?

While this has not been explicitly studied, we are also concerned that large deviation in N and L should not be assessed: *"Note that (35) describes a condition that is assumed to be valid in addition to other changes affecting L and N. Since (35) is only valid after sufficient time has elapsed (18h) (Chen et al., 2024a), stratocumulus that exhibit faster changes in L and N should not be assessed using (35). This might be the case for stratocumulus that are far from their steady state L (Hoffmann et al., 200 2020; Glassmeier et al., 2021; Hoffmann et al., 2024b)."*

It is not explicitly stated whether the key assumption of a zero entrainment response to changes in N to minimum and maximum values applied in the study have been tested with the LES, for example for some extreme values of N, say 10 and 1000 cm-3, and for L = 10 or 1000 g/m2? Perhaps the reader could be directed to relevant sections in the accompanying paper.

The assumption (35) has only been applied to liquid water paths between 10 and 100 g m$^{-2}$ and droplet concentrations between 100 cm$^{-3}$ and 1000 cm$^{-3}$ *"Figure 3 shows $m \equiv d \ln (L)/d \ln (N )$ as a function of N for two different L values: L = 100 g m−2 (continuous lines) is representative for most stratocumulus (e.g., Wood, 2012). L=10 g m$^{-2}$ (dashed lines) is representative for optically thin stratocumulus, which reflect markedly less solar radiation, but cover substantial regions of the globe (e.g., Leahy et al., 2012). Moreover, the emission of longwave radiation for L = 10 g*

*m⁻² is not saturated, enabling the potential for different cloud water adjustments than at higher L. In the following, the case analyzed above (the default) is indicated by black lines, while altered setups are indicated by colored lines. For clarity, we will focus our discussion on how m changes between N = 100 and 1000 cm⁻³, typical values that circumscribe weakly and non-precipitating stratocumulus"* While these values slightly exceed the range of values tested in our companion paper (Chen et al. 2024a), in which liquid water paths in the range of 40 to 100 g m⁻² and droplet concentrations between 100 and 400 cm⁻³ are analyzed (see their Fig. 1), the assumption (35) is probably valid for the largest part of the phase space shown in Fig. 3.

The assumption of constant entrainment is special, as other perturbations, such as changes in surface temperature or free tropospheric conditions, would likely lead to a nonlinear entrainment response. For instance, De Roode et al. (2014) examined the entrainment response to changes in large-scale conditions and found that the entrainment response significantly altered the feedback strength.

While it is interesting to look at other perturbations, those are not in the scope of this study. This study aims to develop an understanding of the processes determining the slope of cloud-water adjustments in weakly and non-precipitating stratocumulus, i.e., feedbacks caused by perturbations in the cloud droplet concentration, which are believed to be primarily caused by a response in the entrainment rate (e.g., Hoffmann et al. 2020). Please note that we investigate how the free-tropospheric moisture effects cloud-water adjustments in Section 3.3.

Additionally, numerous studies using LES models show that changing cloud droplet concentration has a strong impact on entrainment and I am not sure whether those results are in line with the assumption of w_e(N,LWP)=w_e(N+dN,L+dL). The implications of the latter condition warrant a more critical discussion.

We agree that an aerosol perturbation results in an initial increase in the entrainment rate, as stated in our manuscript: *"The large-eddy simulations in our companion paper (Chen et al., 2024a) show that a positive perturbation of N , δN > 0, results in an increase in w_e in response to an aerosol perturbation (see their Fig. 4a). After sufficient time (18 h), this increase in w_e is diminished, resulting in negligible differences in w_e among the perturbed and unperturbed simulations."* Moreover, we state that *"This study was built upon the assumption that an increase in entrainment rate w_e due to an increase in N is exactly offset by a commensurate decrease in L, resulting in the same w_e irrespective of N. This idea is based on large-eddy simulations presented in our companion paper (Chen et al., 2024a) (see their Fig.4a), and can be considered a corollary of the w_e-L feedback mechanism suggested by Zhu et al. (2005) (see their Fig. 7)."* This mechanism has been highlighted by Wood (2012) as being responsible for the observed stability of stratocumulus (see their Fig. 26). Thus, our assumption (35) is not only in line with the large-eddy simulations presented in our companion paper, but also based on our general understanding of stratocumulus-topped boundary layers.

It is difficult to read for non-experts. It could help to start with stating upfront that you will apply a summation of the individual fluxes. An explanation of the MLM, its setup and some of its results in a figure would be helpful, for example like Fig. 11 from the MLM paper by Nicholls (1984) or Figure 4 by Bretherton and Wyant (1997). As a reason, I am not able to derive

whether the total fluxes of the conserved variables are linear in height? Actually, one would expect them to be constant with height, as this implies a steady state.

We do not use a mixed layer model. We only used the entrainment rate parameterization frequently used in mixed layer models, as stated in the abstract: "In this study, we utilize a simple entrainment parameterization used in mixed-layer models to determine entrainment-mediated cloud-water adjustments in non-precipitating stratocumulus." In 2.3, we further clarify that "Using the mixed-layer model entrainment parameterization $w_e$ outlined above, (35) is solved iteratively for $\delta L$ using prescribed values of L, N , and $\delta N$ , while keeping all other parameters constant (e.g., surface fluxes, boundary layer depth)." To use the entrainment rate parameterization, we have to determine the integrated buoyancy flux, as outline in Section 2.2.2, with some ideas related to mixed layer modeling. These calculations are based on existing literature, extensively cited in Section 2.2.2.

"Under well-mixed conditions, contributions to the buoyancy flux that originate from the surface or the top of the boundary layer can be assumed to increase or decrease linearly within the boundary layer". Note that linear flux profiles are only valid for moist conserved thermodynamic variables. This is now stated only implicitly. If the fluxes of conserved variables are linear with height this means that the shape of the vertical profile of the mean state is constant in time. Well-mixedness is not a necessary constraint here: d/dz d/dt X_mean = d/dt d/dz X_mean = - d/dz d/dz w'X' = 0 for a linear flux profiles (the term in the middle indicates the shape of the vertical profile is constant in time).

Our assumption (35) does not require the mixed layer to be in a steady state. (Although we assume the boundary layer depth and other parameters to be constant to simplify our calculations.)

The discussion on optically thin clouds is interesting. I would like to mention the works of Stephens (1978) or Slingo et al. (1982) who showed the relation between the LWP and the downwards emissivity being discussed in the study. It would also be nice to mention that this regime is commonly present, which would strengthen the discussion, for example Leahy et al. (On the nature and extent of optically thin marine low clouds, JGR, 2012).

We have amended the following statement: *"Figure 3 shows m ≡ d ln (L)/d ln (N ) as a function of N for two different L values: L = 100 g m$^{-2}$ (continuous lines) is representative for most stratocumulus (e.g., Wood, 2012). L=10 g m$^{-2}$ (dashed lines) is representative for optically thin stratocumulus, which reflect markedly less solar radiation, but cover substantial regions of the globe (e.g., Leahy et al., 2012). Moreover, the emission of longwave radiation for L = 10 g m$^{-2}$ is not saturated, enabling the potential for different cloud water adjustments than at higher L."* We thank the reviewer for reminding us about Stephens (1978), which has been added the revised manuscript: *"[…] where the first term on the right-hand side describes the radiative cooling across the boundary layer, which is scaled by [1 − exp (−κr L)] to consider the saturation of longwave radiative cooling toward larger L (e.g., Stephens, 1978)."*

One additional issue to consider is that thin clouds may appear broken, in which case the assumptions of the MLM may break down. This possibility should be discussed further.

Broken stratocumulus are typically associated with decoupling, which is discussed in the manuscript: *"The stippling marks potentially decoupled boundary layers, where the buoyancy flux is too weak to ensure a well-mixed boundary layer. These regions have been determined using the approach by Turton and Nicholls (1987). Reasons for the decoupling will be discussed more deeply when addressing ⟨B⟩. As decoupled boundary layers violate many assumptions reasonable for well-mixed boundary layers, this part of the phase space should not be assessed."*

Deardorff's entrainment relation (7) parameterization. Note that Van Zanten et al. (1999) write about Deardorff's (1976) constant: "The value of Add1 is not constant (we found an order of magnitude variation) with respect to all types of convective boundary layers, so the closure is rejected.". Perhaps the equation that introduced the factor k* can be omitted as it is not applied directly in the study, but instead the other entrainment efficiency factor A, which is not constant (Eq. 25).

We have adapted the text as follows: *"Typically, $w_*^3$ is related to the vertically integrated buoyancy flux $h_t\langle B\rangle$ [...] such that $w_*^3 = A\, h_t\langle B\rangle$, with an efficiency factor A (Deardorff, 1976; vanZanten et al., 1999)."* Followed by minor adjustments around (25).

A weak point of the study is that solar radiation is not taken into account. Solar radiation  constitutes a key radiative forcing. The absorption of solar radiation strongly impacts cloud dynamics, as it reduces the effect of longwave radiative cooling. I can imagine that it may have an impact on the results.

We agree that solar radiation has an important impact on the development of stratocumulus, as we have studied previously (Chen et al., 2024b). However, considering solar radiation and its diurnal cycle will cause a time dependency in (35), which for now would complicate our main intent to understand cloud water adjustments. We now state explicitly that *"Similar to our companion paper (Chen et al., 2024a), we neglect interactions with solar radiation for simplicity."* and *"Additionally, (35) does not consider adjustments in surface fluxes (Bretherton and Wyant, 1997), as well as the impact of shortwave radiation (Chen et al., 2024b), for which more complex models would be required."*

Fig. 3c analyzes the influence of free tropospheric humidity on the feedback factor m by halving or doubling the value of the humidity jump across the inversion. Line 375 summarizes the findings:  "Further, we showed that increasing free-tropospheric humidity strengthens negative cloud-water adjustments, in contrast to modeling by Glassmeier et al. (2021), but in agreement with Chun et al. (2023) and our companion paper (Chen et al., 2024a)." Dussen et al. (2015) also found that the inversion humidity jump strongly affects the (equilibrium) LWP and inversion height. With respect to the latter I have difficulties to understand how the experiment in the study was set-up. Is it just a matter of changing the humidity jump Dqt in the entrainment parameterization while keeping the rest the same?

Exactly. As we are only working with assumption (35) and hence only the entrainment parameterization, the humidity jump and the boundary layer depth are changed in the entrainment parameterization while keeping all other parameters the same. We now state: *"To address this, we vary $h_t$ (red and green lines) and $\Delta q_t$ (blue and orange lines) by halving or doubling their default values in (35), while keeping all other parameters the same."* The

reference to Dussen et al. (2015) is very interesting. However, that study assumes a constant droplet concentration and therefore does not enable any insights on the cloud water adjustments studied here.

De Roode, S. R., Siebesma, A. P., Dal Gesso, S., Jonker, H. J., Schalkwijk, J., & Sival, J. (2014). A mixed-layer model study of the stratocumulus response to changes in large-scale conditions. Journal of Advances in Modeling Earth Systems, 6(4), 1256-1270.

Van der Dussen, J. J., S. R. De Roode, S. Dal Gesso, and A. P. Siebesma. "An LES model study of the influence of the free tropospheric thermodynamic conditions on the stratocumulus response to a climate perturbation." Journal of Advances in Modeling Earth Systems 7, no. 2 (2015): 670-691

Leahy, L. V., R. Wood, R. J. Charlson, C. A. Hostetler, R. R. Rogers, M. A. Vaughan, and D. M. Winker. "On the nature and extent of optically thin marine low clouds." Journal of Geophysical Research: Atmospheres 117, no. D22 (2012).

Vanzanten, M. C., Duynkerke, P. G., & Cuijpers, J. W. (1999). Entrainment parameterization in convective boundary layers. Journal of the atmospheric sciences, 56(6), 813-828.

---

## Author Comment (AC2)

Response to Reviewer 2

Summary:

In this paper, the author's examine the relationship between liquid water path (L) and cloud droplet number concentration (N) in the DYCOMS-II nocturnal stratocumulus case using an entrainment parameterization from Deardorff (1976), a partitioning of the buoyancy flux profile into contributions from surface fluxes, entrainment fluxes, long-wave radiation, and precipitation, and a slope parameter (m = dlog(L)/dlog(N)). The mixed-layer model assumes well-mixed boundary layers; therefore, the analysis is restricted to non-precipitating or weakly precipitating profiles. The mixed-layer analysis finds that at low N, the major driver of negative L adjustments is precipitation (virga). At high N, decreased sedimentation velocities increase droplet residence times near cloud top and result in greater entrainment and negative L. At low L (less than 30 g/m$^2$), an increase in droplet size can cause negative L through more efficient long-wave cooling (which enhances entrainment). Overall, L adjustments tend to be strongly negative for N = 10-100 cm$^{-3}$ and tail off to smaller values at very high N (1000 cm$^{-3}$), which differs from the steady-state m in Glassmeier et al. (2021).

We thank the reviewer for the careful reading of our manuscript and the insightful comments made below. Our answers are stated in green font, while excerpts from the (revised) manuscript are printed in blue font with italic characters. All references not explicitly stated in this letter can be obtained from the revised manuscript.

Main comments:

The writing is thorough and technical, but at times hard to digest. I would suggest a title change, as one of the main results of the paper pertains to the effects of precipitating condensate on the mixed-layer state.

We changed the title to *"On the Processes Determining the Slope of Cloud-Water Adjustments in Weakly and Non-Precipitating Stratocumulus"*, with subsequent changes throughout the document.

I find reduced-order approaches such as the MLM outlined in this paper to be attractive and helpful in teasing out causality, but I question just how many "real" environments the results may apply to. If I am not mistaken, all of the L-N phase space is explored from a single research flight with fixed surface fluxes, moisture/energy jumps, and boundary layer depth. Some of these imposed environments may be very unlikely to occur in nature.

The reviewer is right that we present a very large phase space in Figs. 1 and 2, with combinations that are probably not realistic. However, the core of our study is Fig. 3 and the associated discussion in Section 3.3. Here, liquid water paths between 10 and 100 g m$^{-2}$ and droplet concentrations between 100 cm$^{-3}$ and 1000 cm$^{-3}$ are tested, i.e., values typically observed in nature, as we clarified in the revised manuscript: *"Figure 3 shows m ≡ d ln (L)/d ln (N ) as a function of N for two different L values: L = 100 g m$^{-2}$ (continuous lines) is representative for most stratocumulus (e.g., Wood, 2012). L=10 g m$^{-2}$ (dashed lines) is representative for optically thin stratocumulus, which reflect markedly less solar radiation, but cover substantial regions of the globe (e.g., Leahy et al., 2012)."* As we agree that liquid water

paths substantially higher than 100 g m$^{-2}$ exceed the typical range of observed stratocumulus, we added the following statement: *"In fact, all L>150gm−2 are rather untypical for stratocumulus (Wood, 2012), and results in this part of the phase space should be analyzed with care."*

It would be beneficial to the reader to see some connection between the DYCOMS-II case (perhaps even a few LES cases probing the L-N phase space) and MLM results. For example, how do entrainment rates in the MLM compare to the field campaign (or buoyancy flux profiles from the LES)? It would be nice to see some exercise to constrain the behavior of the MLM and describe differences between the idealized and observed profiles.

We have added a black rectangle to Figs. 1 and 2 indicating the L-N range observed during the second research flight of the DYCOMS-II campaign, as reported by Ackerman et al. (2009). Our entrainment rate in this region (6 to 8 mm s$^{-1}$) agrees very well with the observed range of values (Fig. 1a). Moreover, the integrated buoyancy flux (Fig. 1d) or the surface precipitation rate also agree well with the large-eddy simulation intercomparison values reported in Ackerman et al. (2009). We added the following statement to Sec. 2.4: *"Figures 1 and 2 contain black rectangles that indicate the L-N range observed during the second research flight of the DYCOMS-II campaign (Ackerman et al. 2009). The $w_e$ determined from (8) agrees very well with the observed range of 6 to 8 mm s$^{-1}$ (cf. Ackerman et al., 2009). Additionally, ⟨B⟩ (Fig. 1d) and the surface precipitation rate $P_p(0)$ (dashed lines in Figs. 1 and 2) are in general agreement with the values reported from the large-eddy simulation intercomparison by Ackerman et al. (2009). This indicates that the framework applied in this study reproduces the behavior of observed stratocumulus at least in some parts of the analyzed phase space."*

It is stated within the conclusions that "This study was built upon the assumption that an increase in entrainment rate we due to an increase in N is exactly offset by a commensurate decrease in L, resulting in the same we irrespective of N." This statement is at odds with Figure 1a, with the entrainment rate contours sloping with changing N.

Entrainment increases with N. At several places, we state that *"In the absence of precipitation, further increases in N are found to increase the mixing of clouds with their surroundings (entrainment), leading to a decrease in L (Wang et al., 2003; Ackerman et al., 2004; Bretherton et al., 2007)."* and *"The large-eddy simulations in our companion paper (Chen et al., 2024a) show that a positive perturbation of N , δN > 0, results in an increase in we in response to an aerosol perturbation (see their Fig. 4a)."* However, this increase is transient because entrainment introduces warm and dry free-tropospheric air into the boundary layer, which evaporates the cloud. As a result, L decreases. And since entrainment is also a function of L, an initial increase in the entrainment rate due to a perturbation in N is offset by a (negative) adjustment in L: *"After sufficient time (18 h), this increase in $w_e$ is diminished, resulting in negligible differences in $w_e$ among the perturbed and unperturbed simulations. This decrease in $w_e$ is enabled by a commensurate decrease in L in the perturbed simulations, δL < 0, resulting in increasingly stronger negative m with time (see their Fig. 2c)."* While the assumption (35) is an idealization of this behavior, it is based on accepted theories. For instance, this idealization can be considered a corollary of the feedback mechanism between entrainment rate and liquid water path originally suggested by Zhu et al. (2005) (see their Fig. 7), which has been also

highlighted by Wood (2012) as a major mechanism responsible for the observed stability of stratocumulus (see their Fig. 26). We state this as: *"This idea is based on large-eddy simulations presented in our companion paper (Chen et al., 2024a) (see their Fig.4a), and can be considered a corollary of the $w_e$-L feedback mechanism suggested by Zhu et al. (2005) (see their Fig. 7)."*

Additionally, it is not clear from Chen et al. (2024a) that this assumption is justified. In Chen et al. (2024a), the L-N phase space (L ranges from 65-95 g/m$^2$; N ranges from 100-600 cm$^{-3}$) is much smaller than the phase space examined in this paper. Given how critical this assumption remains to the results of the paper, justification for this simplification needs to be provided within the manuscript.

While we show a much greater phase in Figs. 1 and 2, these results do not rely on the assumption (35). These results are based on the established entrainment parameterization discussed in Sections 2.2. There is nothing new about the entrainment parameterization, besides showing how the entrainment parameterization and its components behave in an L-N phase space, which is necessary to better understand Fig. 3.

Only for Fig. 3 and the associated discussions in Section 3.3, has the assumption (35) been applied to liquid water paths between 10 and 100 g m$^{-2}$ and droplet concentrations between 100 cm$^{-3}$ and 1000 cm$^{-3}$: *"Figure 3 shows $m \equiv d \ln (L)/d \ln (N)$ as a function of N for L = 10 and 100 g m$^{-2}$ (dashed and continuous lines, respectively), representing cases with unsaturated and saturated longwave radiative cooling. [...] For clarity, we will focus our discussion on how m changes between N = 100 and 1000 cm$^{-3}$, typical values that circumscribe non-precipitating stratocumulus."* While these values slightly exceed the range of values tested in our companion paper (Chen et al. 2024a), in which liquid water paths in the range of 40 to 100 g m$^{-2}$ and droplet concentrations between 100 and 400 cm$^{-3}$ are analyzed (see their Fig. 1), the assumption (35) is probably valid for the largest part of the phase space shown in Fig. 3.

It's also unclear why this assumption is necessary. In Section 2.3.2, the timescale of 18 hours to reach an equilibrium entrainment rate is mentioned, but this would require a longer time period than a single night to equilibrate.

We agree that 18 hours exceed the nighttime in the subtropics where most stratocumulus are found. However, for this study and many previous studies on aerosol-cloud interactions (e.g., Glassmeier et al. 2021) or our companion paper (Chen et al. 2024a), we assume perpetual night to better understand the interactions of cloud microphysics and dynamics. As soon as solar radiation is considered, the interpretation of results is much harder. We clarify that we do not consider solar radiation: *"As for our companion paper (Chen et al., 2024a), we neglect interactions with solar radiation for simplicity."* and *"Additionally, (35) does not consider adjustments in surface fluxes (Bretherton and Wyant, 1997), as well as the impact of solar radiation (Chen et al., 2024b), for which more complex models would be required."*

Please explain why entrainment cannot vary as a function of N.

See our previous answer to "It is stated within the conclusions that "This study was built upon the assumption that an increase in entrainment rate we due to an increase in N is exactly offset by a commensurate decrease in L, resulting in the same we irrespective of N." This statement is at odds with Figure 1a, with the entrainment rate contours sloping with changing N.".

In addition to the entrainment assumption, it seems odd to fix the boundary layer depth in the MLM. Is there a reason the moisture and energy equations are decoupled from the mass budget?

There are no coupled or decoupled budgets used in this study. The only equation evaluated in this study is (8): *"In this study, we utilize a simple entrainment parameterization used in mixed-layer models to determine entrainment- mediated cloud-water adjustments in non-precipitating stratocumulus."* In 2.3, we further clarify that *"Using the mixed-layer model entrainment parameterization $w_e$ outlined above, (35) is solved iteratively for $\delta L$ using prescribed values of L, N , and $\delta N$ , while keeping all other parameters constant."* One could address changes in boundary layer depth in (35). However, this would require knowledge on how the boundary layer depth changes with N. While this is interesting, this study aims to provide deeper understanding of cloud water adjustments, i.e., the change in L with N. While the boundary layer depth certainly has an impact on the cloud water adjustments shown in Fig. 3c, the change in boundary layer depth over the course of 18 h, as implicitly assumed for (35), is probably small in contrast to the more direct impact of entrainment on L. Still, investigating these nuances is important, and we amended our last to paragraph accordingly: *"All in all, this study showed that even comparably simple models, such as the one used here, can be applied to increase our fundamental understanding of aerosol-cloud interactions. In fact, the simplicity of the applied model allowed us to directly link cause and effect of cloud-water adjustments, which can be difficult in more complex models such as global circulation or even large-eddy simulation models due to confounding factors (Mülmenstädt et al., 2024).. That being said, the nuance provided by these models should not be disregarded as they help to quantify the effects that have been neglected here (e.g., interactive surface fluxes, changes in boundary layer depth, or the diurnal cycle of solar radiation). Thus, we advocate that the assessment of aerosol-cloud interactions should balance the use of complex and simple approaches by substantiating quantitative understanding with qualitative insight."*

Minor comments:

1. Line 6: "At higher N, the cessation"... here and on line 370, I would recommend not using "cessation", as sedimentation continues to exist, it is just smaller.

   Changed to *"decrease in"*.

2. Line 23: It may be useful to mention the diurnal cycle of LWP (Wood et al., 2002). Recent LES of polluted environments have shown larger diurnal variability of LWP (at nearly constant N). It's important to consider how the diurnal cycle of entrainment rate/buoyancy flux may influence the interpretation of m. Additionally, it may be worth noting that the free troposphere can act as a source of N, complicating a clean relationship between the entrainment rate and the liquid water response (Chun et al., 2023).

   Those processes confound the understanding that can be gained from observations. Thus, disregarding them in our simplified framework helps to reveal the underlying physics more clearly. We have added the following statement to the manuscript's introduction: *"Additionally, observational approaches to estimate m need to consider the natural*

*variability in L due to, e.g., the diurnal cycle of solar radiation (e.g., Wood et al., 2002) or changes in the large-scale meteorological conditions (e.g., Chun et al., 2023; Goren et al., 2024). While models can be used to eliminate the influence of some of these confounding factors, the variability of m in weakly or non-precipitating clouds is likely associated with a complex network of interactions and dependencies that comprise entrainment (Mellado, 2017; Igel, 2024), making it hard to obtain direct process understanding from models that represent the underlying dynamics explicitly, e.g., three-dimensional large-eddy simulations. Thus, to understand m for weakly and non-precipitating stratocumulus better, we will base our work on a simple, zero-dimensional mixed-layer model (Lilly, 1968; Schubert et al., 1979; Bretherton and Wyant, 1997; Stevens, 2002). We will focus on the representation of the entrainment rate in such models and how it depends on L and N , which allows us to directly assess the individual aspects of the entrainment process (Nicholls and Turton, 1986; Turton and Nicholls, 1987; Bretherton et al., 2007)."*

3. Line 55: The boundary-layer depth mass budget can also depend on horizontal advection of ht (Caldwell et al., 2005).

   This probably depends on whether one considers the simulated volume of air in a Lagrangian framework or at a predefined location. We added the following statement: *"Lastly, s, qt, and ht can change due to large-scale advection."*

4. Line 217: This may need to be quantified. What value of negative cloud-base buoyancy flux was used to determine decoupling?

   We have added the following statement: *"These regions have been determined using the approach by Turton and Nicholls (1987), which diagnoses decoupling if the ratio of the integrated sub-cloud-layer buoyancy flux to the integrated cloud-layer buoyancy flux is smaller than −0.4."*

5. Line 226: Why is a cloud-base precipitation rate necessary to omit cases if we only care about decoupling? If a precipitating case remains fairly well-mixed (no significant negative cloud-base buoyancy flux) there is no need to get rid of that profile.

   We have clarified this sentence as: *"Since our approach does not consider losses in L due to surface precipitation, we neglect regions of the L-N phase space that are substantially affected by this process."*

6. Line 232: A decrease in the buoyancy jump does not necessarily guarantee an increase in entrainment rate.

   This is true, and is generally considered in the entrainment rate parameterization (8) and its components. Here, however, we are considering the general behavior of $\Delta b$ with L as represented by (11), and not the entrainment rate directly.

Stronger buoyancy jumps can be created by more effective long-wave cooling (McMichael et al., 2019). Also, low cloud fraction environments may have diffuse inversion layers but reduced entrainment in the absence of radiatively- and evaporatively-driven turbulence.

These interactions are considered in (11) via $\Delta s$ and $\Delta q_t$. However, these are assumed constant for Figs. 1 and 2. The influence of $\Delta q_t$ is discussed in Sec. 3.3.

7.  Line 270: Negative buoyancy near cloud base may not guarantee a decrease in entrainment, but may aid in decoupling. Turbulence production near cloud-top is the more critical quantity.

    As indicated by (5) to (8) and many detailed studies on this subject (e.g., Yamaguchi and Randall 2008), entrainment is not determined by (the small-scale) turbulence production at the cloud-top but the boundary layer's large-scale dynamics. Thus, changes in the integrated buoyancy flux $\langle B \rangle$ primarily determine the entrainment rate.

8.  Lines 336-341: This paragraph makes it seem like the diurnal cycle was taken into consideration. I would recommend emphasizing the diurnal uncertainties that still exist in m.

    The interaction of clouds with solar radiation is probably the strongest motivation to study aerosol-cloud interactions. Not mentioning solar radiation in the summary/conclusion feels sacrilegious. To clarify that solar radiation has been neglected for our study, we amended the following statement: *"[…] Still, the additional nuance provided by these models should not be disregarded, as they help quantifying the effects that have been neglected here (e.g., interactive surface fluxes, changes in boundary layer depth, or the diurnal cycle of shortwave radiation). Thus, we advocate that the assessment of aerosol-cloud interactions should balance the use of complex and simple approaches by substantiating quantitative understanding with qualitative insight."*

References:

Wood, R., C. S. Bretherton, and D. L. Hartmann, Diurnal cycle of liquid water path over the subtropical and tropical oceans, Geophys. Res. Lett., 29(23), 2092, doi:10.1029/2002GL015371, 2002.

McMichael LA, Mechem DB, Wang S, Wang Q, Kogan L, Teixeira J. Assessing the mechanisms governing the daytime evolution of marine stratocumulus using large-eddy simulation. Q J R Meteorol Soc. 2019; 145: 845–866. https://doi.org/10.1002/qj.3469

Yamaguchi, T., & Randall, D. A. (2008). Large-eddy simulation of evaporatively driven entrainment in cloud-topped mixed layers. *Journal of the Atmospheric Sciences*, *65*(5), 1481-1504.

---

## Referee Report (RR1)

**Reviewer 2 Final Comments**

The statement "This study was built upon the assumption that an increase in entrainment rate $(w_e)$ due to an increase in $N$ is exactly offset by a commensurate decrease in $L$, resulting in the same $w_e$ irrespective of N" has the following implication:

$$\frac{dw_e}{dN} = 0, \tag{1}$$

with the chain rule expansion resulting in

$$\frac{dw_e}{dN} = \frac{\partial w_e}{\partial N} + \frac{\partial w_e}{\partial L}\frac{dL}{dN} = 0. \tag{2}$$

Rearranging the above equation gives

$$\frac{dL}{dN} = -\left(\frac{\partial w_e}{\partial N} \Big/ \frac{\partial w_e}{\partial L}\right). \tag{3}$$

Eq. 3 then imposes the following behavior on the slope parameter $(m)$,

$$m = \frac{dln(L)}{dln(N)} = \frac{N}{L}\frac{dL}{dN} < 0. \tag{4}$$

This assumption results in a strong constraint on the slope parameter that vastly decreases the utility of the analysis. Entrainment rate would be expected to change as a function of $N$, and the assumption that $L$ exactly and immediately compensates any $N$-related entrainment changes is questionable, given that the companion study required nearly 18 hours to equilibrate, which is far more than a single nighttime period. Clouds in nature would likely never have time to fully equilibrate overnight before diurnally varying solar insolation muddies the picture. The dependence of Eq. 1 on there being a perpetual night calls the entire study into question. Additionally, neglecting cloud deepening/thinning, fixed surface fluxes, and restricting the analysis to nighttime conditions are fundamental weaknesses of the study that results in decreased relevance to any real world scenario. While the authors mention that adding complexity would complicate the analysis, there are many ways to account for these processes within the MLM framework using moisture, energy, and mass budgets. One of the three main results pertains to the evaporation of precipitation, which introduces an internal inconsistency into the study. Virga decreasing $w_e$ is not compatible with Eq. (1), since this complicates the direct relationship between $N$, $L$, and $w_e$ and would allow for Eq. 4 to be positive in precipitating cases. It is not clear from Fig. 1 and 2 which cases have virga and which ones don't. This would be a helpful addition to the plot and help determine just how big of an issue the virga-mediated entrainment feedback may be.

**Overall, the extent of constraining assumptions does not allow for realistic responses and the results are only applicable in an extremely narrow set of (unlikely/unnatural) circumstances. I don't believe the results significantly advance our previous understanding of sedimentation-entrainment feedbacks and I believe that the virga results contradict the assumption in Eq. 1 and could theoretically result in positive $m$.**

---

## Author Response (AR2)

Response to Editor

Please respond to the final comments made by reviewer #2. They raise legitimate concerns regarding the rather stringent conditions considered in this study. I believe that you can address these concerns with additional caveats added to the paper. In particular, the abstract should identify the idealized steady state nature of the simulations, particular focus on coupled boundary layers, and emphasize that precipitating conditions were not considered.

We thank the editor for the additional comments. Our answers to Reviewer 2 can be found below. We added an additional statement to the abstract to revised manuscript to highlight the idealized nature of our study: *"Using this idealized framework that neglects interactive surface fluxes, changes in boundary layer depth, or the diurnal cycle of solar radiation, we are able to show that cloud-water adjustments weaken distinctly from dln(L)/dln(N)=-0.48 at N=100 cm$^{-3}$ to -0.03 at N=1000 cm$^{-3}$, indicating that a single value to describe cloud-water adjustments in weakly and non-precipitating clouds is insufficient."*

Response to Reviewer 2

We thank the reviewer for the additional comments on our manuscript.

My main major concern remains: the extent of constraining assumptions does not allow for realistic responses and the results are only applicable in an extremely narrow set of circumstances. I explain my thoughts a bit more deeply in the PDF and provide some mathematical explanations for why I feel this why. Instead of focusing solely on the entrainment rate treatment in the mixed layer model context, I think the study would benefit from the use of a full MLM, which would broaden the scope of the results to include conditions that may be more common in nature.

We agree with the reviewer that this study is idealized. However, this study's primary goal is to increase our conceptual understanding of negative cloud-water adjustments in weakly and non-precipitating stratocumulus. Previously, most studies showed a large variety of constant slopes to describe negative cloud-water adjustments (e.g., Fig. 1 in Glassmeier et al. 2021). Our study clearly shows that the slope is a function of the droplet concentration and might explain why literature values differ so much. As stated before, this study does not aim for an exact quantitative description of the underlying physics, but a better qualitative understanding. As suggested by the editor, we now emphasize the idealized nature of our study also in the abstract: *"Using this idealized framework that neglects interactive surface fluxes, changes in boundary layer depth, or the diurnal cycle of solar radiation, we are able to show that cloud-water adjustments weaken distinctly from dln(L)/dln(N)=-0.48 at N=100 cm$^{-3}$ to -0.03 at N=1000 cm$^{-3}$, indicating that a single value to describe cloud-water adjustments in weakly and non-precipitating clouds is insufficient."*

The statement "This study was built upon the assumption that an increase in entrainment rate (we) due to an increase in N is exactly offset by a commensurate decrease in L, resulting in the same we irrespective of N" has the following implication:

$$\frac{dw_e}{dN} = 0, \tag{1}$$

with the chain rule expansion resulting in

$$\frac{dw_e}{dN} = \frac{\partial w_e}{\partial N} + \frac{\partial w_e}{\partial L}\frac{dL}{dN} = 0. \tag{2}$$

Rearranging the above equation gives

$$\frac{dL}{dN} = -\left(\frac{\partial w_e/\partial N}{\partial w_e/\partial L}\right). \tag{3}$$

Eq. 3 then imposes the following behavior on the slope parameter (m),

$$m = \frac{dln(L)}{dln(N)} = \frac{N}{L}\frac{dL}{dN} < 0. \tag{4}$$

This assumption results in a strong constraint on the slope parameter that vastly decreases the utility of the analysis.

We agree with this analysis. However, the negative cloud-water adjustments that we postulate are based on many studies on aerosol-cloud interactions that have shown that this entrainment-related effect exists (e.g., Fig. 1 in Glassmeier et al. 2021). Our study wants to show what conclusions can be drawn when using it as a starting point.

Entrainment rate would be expected to change as a function of N, and the assumption that L exactly and immediately compensates any N-related entrainment changes is questionable, given that the companion study required nearly 18 hours to equilibrate, which is far more than a single nighttime period. Clouds in nature would likely never have time to fully equilibrate overnight before diurnally varying solar insolation muddies the picture. The dependence of Eq. 1 on there being a perpetual night calls the entire study into question. Additionally, neglecting cloud deepening/thinning, fixed surface fluxes, and restricting the analysis to nighttime conditions are fundamental weaknesses of the study that results in decreased relevance to any real world scenario. While the authors mention that adding complexity would complicate the analysis, there are many ways to account for these processes within the MLM framework using moisture, energy, and mass budgets. One of the three main results pertains to the evaporation of precipitation, which introduces an internal inconsistency into the study. Virga decreasing we is not compatible with Eq. (1), since this complicates the direct relationship between N, L, and we and would allow for Eq. 4 to be positive in precipitating cases. It is not clear from Fig. 1 and 2 which cases have virga and which ones don't. This would be a helpful addition to the plot and help determine just how big of an issue the virga-mediated entrainment feedback may be.

Again, we agree with these thoughts for the most part. However, we would like to reiterate that we do not believe that the evaporating virga creates an external inconsistency, as we made sure that losses of cloud water to the surface are negligible: *"Since our approach does not consider losses in L due to surface precipitation, we neglect regions of the L-N phase space that are substantially affected by this process. The dashed line marks the surface precipitation rate $P_p(0)=0.1\ w'q_t'|_0\ \varrho_0 = 0.3\ mm\ d^{-1}$. By determining this value based on $w'q_t'|_0$, we identify regions of the L-N phase space in which precipitation losses are substantial (to the left of the dashed line) and the region where precipitation losses are negligible, i.e., the weakly and non-precipitating clouds that are the main focus of this study (to the right of the dashed line)."* The effect of virga can be easily determined from Fig. 2e, which clearly shows the extent of the analyzed phase space (right of the dashed line) that is affected by virga. By deactivating the

influence of virga (green continuous line), Fig. 3a clearly shows what part of the phase space is affected.

Overall, the extent of constraining assumptions does not allow for realistic responses and the re- sults are only applicable in an extremely narrow set of (unlikely/unnatural) circumstances. I don't believe the results significantly advance our previous understanding of sedimentation-entrainment feedbacks and I believe that the virga results contradict the assumption in Eq. 1 and could theo- retically result in positive m.

As stated above, the virga considered here does not substantially affect the cloud-water budget. We agree that stronger precipitation results in positive cloud-water adjustments, as previously shown by Albrecht (1989) and others, and usually referred to as "precipitation suppression", where an increase in N results in smaller cloud droplets that do not collide as frequently and hence produce less precipitation, that would remove cloud water from the system through surface losses. By focusing on virga, in which only negligible amounts of cloud water reach the surface because it almost completely evaporates below cloud base, the total water of the system is only marginally affected by this form of precipitation. However, our study clearly shows that the effect of the evaporating virga has implications for the buoyancy produced by the boundary layer (Figs. 1d, 2e, and 3a). To our knowledge, this effect has not been fully recognized in the community's discussion on cloud-water adjustments. Thus, our study should be seen as an incentive for future work on this topic, as we cannot answer all questions on this issue due to its idealized nature. In the last version of the manuscript, we highlighted the idealized nature of our study in several locations: *"All in all, this study showed that even comparably simple models, such as the one used here, can be applied to increase our fundamental understanding of aerosol-cloud interactions. In fact, the simplicity of the applied model allowed us to directly link cause and effect of cloud-water adjustments, which can be difficult in more complex models such as global circulation or even large-eddy simulation models due to confounding factors (Mülmenstädt et al., 2024). That being said, the nuance provided by these models should not be disregarded as they help to quantify the effects that have been neglected here (e.g., interactive surface fluxes, changes in boundary layer depth, or the diurnal cycle of solar radiation). Thus, we advocate that the assessment of aerosol-cloud interactions should balance the use of complex and simple approaches by substantiating quantitative understanding with qualitative insight."* and *"Additionally, (35) does not consider adjustments in surface fluxes (Bretherton and Wyant, 1997), as well as the impact of solar radiation (Chen et al., 2024b), for which more complex models would be required. Nonetheless, the large-eddy simulations presented in our companion paper (Chen et al., 2024a) indicate that sufficient physics are captured in (35) to yield reasonable insights, as we will detail below."* In the revised version, we also emphasize this idealized nature in the abstract, *"Using this idealized framework that neglects interactive surface fluxes, changes in boundary layer depth, or the diurnal cycle of solar radiation, we are able to show that cloud-water adjustments weaken distinctly from $dln(L)/dln(N)=-0.48$ at $N=100$ cm$^{-3}$ to $-0.03$ at $N=1000$ cm$^{-3}$, indicating that a single value to describe cloud-water adjustments in weakly and non-precipitating clouds is insufficient."*